# CONFORMALIZED SURVIVAL ANALYSIS FOR GENERAL RIGHT-CENSORED DATA

**Hen Davidov**[†]**, Shai Feldman**[†]**, Gil Shamai**[†]**, Ron Kimmel**[†]**, Yaniv Romano**[†§]

{dahen, shai.feldman}@campus.technion.ac.il,
{sgils, ron}@cs.technion.ac.il,
yromano@technion.ac.il

## ABSTRACT

We develop a framework to quantify predictive uncertainty in survival analysis, providing a reliable lower predictive bound (LPB) for the true, unknown patient survival time. Recently, conformal prediction has been used to construct such valid LPBs for *type-I right-censored data*, with the guarantee that the bound holds with high probability. Crucially, under the type-I setting, the censoring time is observed for all data points. As such, informative LPBs can be constructed by framing the calibration as an estimation task with covariate shift, relying on the conditionally independent censoring assumption. This paper expands the conformal toolbox for survival analysis, with the goal of handling the ubiquitous *general right-censored setting*, in which either the censoring or survival time is observed, but not both. The key challenge here is that the calibration cannot be directly formulated as a covariate shift problem anymore. Yet, we show how to construct LPBs with distribution-free finite-sample guarantees, under the same assumptions as conformal approaches for type-I censored data. Experiments demonstrate the informativeness and validity of our methods in simulated settings and showcase their practical utility using several real-world datasets.

## 1 INTRODUCTION

Survival analysis is essential in numerous fields, including medicine (Cole & Hudgens, 2010; Selvin, 2008), engineering (Ma & Krings, 2008), and social sciences (Cloyes et al., 2010). The primary objective is to predict the survival time $T$—the time-to-event such as death, failure, or relapse—based on covariates $X$ that may include clinical markers, machine specifications, or sociological factors. The underlying challenge in survival analysis is that $T$ is not observed for all subjects due to censoring. In particular, in this paper, we focus on *right-censored data*, the most common setting in survival analysis, where the survival event may not occur by the end of the follow-up period. This means that for some subjects $X$, the true survival time $T$ is obscured by the censoring time $C$. As such, the observed censored survival time is given by $\tilde{T} = \min(T, C)$.

To formalize the above data generation process, we assume the triplets $(X_i, T_i, C_i)$, $i = 1, \ldots, N$, are sampled i.i.d. from $P_{X,T,C}$. The observed data, however, include censored observations, where we consider two possible forms of right-censored survival data.

**type-I right censoring**: Here, we assume that censoring time $C$ is observed for all subjects, which typically occurs in studies with a fixed duration, where all subjects are followed until the study ends. Under this setting, the observed dataset is of the form $\mathcal{D}_{\text{type-I}} = \{(X_i, \tilde{T}_i, C_i)\}_{i=1}^N$.

**General right censoring**: Unlike the type-I setting, in this more general case, we observe either $C$ or $T$ for each subject, but not both. For example, in situations where the censorship time is obscured by early mortality (Candès et al., 2023; Ahmad et al., 2017; Chandrashekar et al., 2014). Concretely, denote by $e_i = \mathbb{I}\{T_i < C_i\}$ a binary indicator variable that gets the value 1 if $T_i$ is observed; that is, $\tilde{T} = T$ when $e = 1$ and $\tilde{T} = C$ otherwise. The observed dataset is then given by

---

[†]Department of Computer Science, Technion – Israel Institute of Technology
[§]Department of Electrical and Computer Engineering, Technion – Israel Institute of Technology

$\mathcal{D}_{\text{general}} = \{(X_i, \tilde{T}_i, e_i)\}_{i=1}^N$. Due to its greater generality, this setup is applicable for many survival analysis problems in biomedical research (Klein, 2003; Cole & Hudgens, 2010) and other fields. Thus, it is the focus of this work.

Given the observed censored data, the learning task is to estimate the distribution of $T \mid X$. However, since we do not fully observe the survival time $T$ for all subjects, we need to impose further assumptions to make the learning task feasible. Indeed, a common assumption in survival analysis is that, conditional on the covariates $X$, the survival time $T$ and the censoring time $C$ are independent. Formally, this is expressed as follows.

**Assumption 1.1** (Conditionally Independent Censoring (Kalbfleisch & Prentice, 2011))**.**

$$C \perp T \mid X.$$

Building on the above assumption and related ones, various machine learning approaches have been successful in estimating the distribution of $T|X$ (Cox, 1992; Wei, 1992; Lee et al., 2018; Nagpal et al., 2021b; Katzman et al., 2018). Despite their notable successes, these methods often lack reliability guarantees for the resulting predictions. This is attributed either to the opaque nature of modern deep learning algorithms or to the simplified modeling assumptions of more traditional statistical methods. Yet, such guarantees are of great importance given the harsh consequences of making erroneous predictions in high-stakes domains, such as healthcare (Navarro et al., 2021; Obermeyer et al., 2019).

### 1.1 THE NEED FOR RELIABLE LOWER PREDICTIVE BOUNDS FOR SURVIVAL ANALYSIS

In response to this challenge, we aim to construct a lower predictive bound (LPB) on the survival time, supporting any given predictive model with distribution-free finite-sample guarantees (Vovk et al., 2005; Vovk, 2012; Tibshirani et al., 2019; Romano et al., 2019; Candès et al., 2023). Our goal is to construct an LPB function $\hat{L}(\cdot)$ such that, for a new test instance $X_{\text{test}}$ with an unknown true survival time $T_{\text{test}}$, the event $\hat{L}(X_{\text{test}}) \leq T_{\text{test}}$ occurs with at least $1 - \alpha$ probability. We refer to the probability of this event as the *coverage rate* of the LPB. For example, if the LPB satisfies the coverage property at level $(1 - \alpha) = 0.9$, then $\hat{L}(X_{\text{test}})$ is guaranteed to be lower than the unknown $T_{\text{test}}$ at least 90% of the time. To further enhance reliability, we seek to obtain a coverage guarantee that holds *conditional on the observed dataset*, in a manner similar to Gui et al. (2024); Park et al. (2019); Bates et al. (2021); Angelopoulos et al. (2021). To this end, let $\delta \in (0, 1)$ be a tolerance level that accounts for the randomness in the realization of the dataset, and define the probably approximately correct (PAC)-type LPB as follows.

**Definition 1.1.** *Let $(X_i, T_i, C_i)$, $i = 1, \ldots, N$ be i.i.d. samples from $P_{X,T,C}$, with $\hat{L}$ being a function of the observed dataset $\mathcal{D} = \mathcal{D}_{\text{general}} = \{(X_i, \tilde{T}_i, e_i)_{i=1}^N\}$, where $\tilde{T}_i = \min(T_i, C_i)$ and $e_i = \mathbb{I}[T_i < C_i]$. $\hat{L}$ is a PAC-type LPB at level $\alpha \in (0, 1)$ with tolerance $\delta \in (0, 1)$ if, with probability at least $1 - \delta$ over the realization of $\mathcal{D}$,*

$$\mathbb{P}\big(T_{\text{test}} \geq \hat{L}(X_{\text{test}}) \mid \mathcal{D}\big) \geq 1 - \alpha,$$

*where the probability is taken with respect to a new data point $(X_{\text{test}}, T_{\text{test}}) \sim P_{X,T}$.*

Recently, conformal prediction methods have been applied to construct valid LPBs for *type-I right-censored data*. The key idea is to use holdout samples to calibrate the predictions of a survival analysis model, ensuring the desired coverage level is obtained for new test points. Under the assumption of conditionally independent censoring, Candès et al. (2023) introduced a conformal method that constructs valid and informative LPBs in finite samples. This approach was further refined by Gui et al. (2024), offering more informative LPBs with PAC-type guarantees. Both methods rely on the availability of the censoring time $C$ for all subjects, which is used to discard early-censored samples. Intuitively, discarding such samples brings the observed time $\tilde{T}$ closer to the true event time $T$, yielding more accurate LPBs. Specifically, leveraging Assumption 1.1, Candès et al. (2023); Gui et al. (2024) reformulate the LPB construction as an estimation problem under covariate shift. Notably, the above calibration methods produce LPBs that are doubly robust, meaning they remain valid if either the predictive model or the estimation of the covariate shift likelihood ratio is accurate.

In the *general right-censored setting*, however, the literature is less developed, and existing methods lack such strong validity guarantees. This is primarily because the censoring time $C$ is not available

for all subjects, making the calibration process more challenging. Qi et al. (2024) suggest imputing the value of $T$ with a 'best guess' to calibrate the estimated survival distribution. Naturally, the validity of the method is affected by the quality of the imputation, and our experiments show that their method often fails to achieve the desired coverage in finite-sample. Concurrently with our work, Qin et al. (2024) and Meixide et al. (2024) proposed distribution-free methods for constructing LPBs in the general right-censored setting, but their bootstrap-based approaches rely solely on asymptotic validity under certain regularity conditions. As a result, finite-sample calibration remains elusive. This is especially concerning because small sample sizes are common in biomedical and psychological research, often undermining the validity and reproducibility of statistical findings (Errington et al., 2021; Nosek et al., 2022; Button et al., 2013; Nüesch et al., 2010).

## 1.2 OUR CONTRIBUTIONS

We build on the foundations of Gui et al. (2024) and propose the first method for constructing *finite-sample* valid LPBs for general right-censored survival analysis data. To produce informative LPBs, we leverage the event indicator $e$ and introduce a novel approach to selectively discard certain censored examples in a data-driven manner, bringing the censored event time $\tilde{T}$ closer to the true event time $T$. *Crucially, although the calibration cannot be directly formulated as a covariate shift problem, our approach forms doubly robust, valid LPBs under the same conditional independence assumption 1.1 of conformal methods for the type-I setting.* Additionally, unlike Meixide et al. (2024); Qi et al. (2024); Qin et al. (2024), our methods' validity relies on accurate estimation of the fully observed $e$, rather than the partially observed $T$ or $C$. We validate our theory and assess the applicability of the proposed methods using synthetic data, and further demonstrate their effectiveness on real breast cancer data. A Python implementation of our methods is provided in our github repository.

## 2 BACKGROUND AND RELATED WORK

Denote by $q_\tau(x)$ the true $\tau$-th quantile of $T \mid X = x$, and define the oracle LPB for $T_{\text{test}}$ as the $\alpha$-th conditional quantile function, i.e., $L(X_{\text{test}}; \alpha) = q_\alpha(X_{\text{test}})$. While this oracle LPB is guaranteed to attain $1 - \alpha$ coverage by definition, in practice we do not have access to the true $q_\alpha(x)$, rendering this approach infeasible. As a way out, one could use survival analysis tools to estimate the conditional quantile function, which we denote by $\hat{q}_\tau(x)$, and form a heuristic LPB for $T_{\text{test}}$ by setting $\hat{L}(X_{\text{test}}; \alpha) = \hat{q}_\alpha(X_{\text{test}})$. This heuristic approach, however, may not attain the desired coverage, unless the quantile estimates are accurate.

In this paper, we alleviate this issue by building on conformal prediction—a general framework to calibrate the estimated LPB. The appeal of conformal prediction is that it allows us to work with any predictive model and guarantee the desired $1 - \alpha$ coverage at test time. The key idea is to modify the LPB function $\hat{L}(x; \tau)$ by rigorously tuning a hyperparameter $\tau \in \mathbb{R}$ using holdout calibration data $\mathcal{I}_{\text{cal}}$ to attain the desired coverage rate. We denote the tuned hyper-parameter as $\hat{\tau}$, and the resulting LPB as $\hat{L}(X; \hat{\tau})$.

### 2.1 FIRST STEPS: NAIVE CONFORMALIZED SURVIVAL ANALYSIS

Before presenting our method, we first outline a naive conformal prediction approach for tuning $\tau$. In addition to using this approach as a baseline method in our experiments, it introduces the core principles of LPB calibration and motivates the need for more powerful solutions.

The key observation here is that $\tilde{T} \leq T$ by definition, and therefore a valid LPB for $\tilde{T}$ is also a valid (but conservative) LPB for $T$. As such, a naive approach to tune $\hat{\tau}$ is to discard the information provided by the event indicator $e_i$ and construct a valid LPB for $\tilde{T}$ using the calibration points $\{(X_i, \tilde{T}_i)\}_{i \in \mathcal{I}_{\text{cal}}}$. In more detail, we define the LPB function as $\hat{L}(X; \tau) = \hat{q}_\tau(X)$, which is tightly connected to the methods by Chernozhukov et al. (2021); Gui et al. (2024).[1] Then, we formulate a (conservative) empirical estimator of the true miscoverage rate $\alpha(\tau) = \mathbb{P}(T < \hat{q}_\tau(X))$ for each

---

[1]There are other possible design choices for $\hat{L}(X; \tau)$, such as the conformalized quantile regression (CQR) approach (Romano et al., 2019) used by Candès et al. (2023).

choice of $\hat{q}_\tau(\cdot)$, $\tau \in [0,1]$, defined as follows:

$$\hat{\alpha}_{\text{naive}}(\tau) = \frac{1}{|\mathcal{I}_{\text{cal}}|} \sum_{i \in \mathcal{I}_{\text{cal}}} \mathbb{I}\left\{\tilde{T}_i < \hat{q}_\tau(X_i)\right\}. \tag{1}$$

Above, we use the indicator function $\mathbb{I}(\cdot)$ to count the events where $\tilde{T}_i < \hat{q}_\tau(X_i)$, and so $\hat{\alpha}_{\text{naive}}(\tau)$ is an unbiased estimator of $\alpha_{\text{naive}}(\tau) = \mathbb{P}(\tilde{T} < \hat{q}_\tau(X))$. Since $\tilde{T} \leq T$ we get that $\alpha_{\text{naive}}(\tau) \geq \alpha(\tau)$, implying that the empirical quantity $\hat{\alpha}_{\text{naive}}(\tau)$ is anticipated to overestimate the miscoverage rate, on average. Finally, we define $\hat{\tau}_{\text{naive}}$ as the smallest value of $\tau$ for which $\hat{\alpha}_{\text{naive}}(\tau)$ meets the user-defined miscoverage level $\alpha$, i.e.,

$$\hat{\tau}_{\text{naive}} = \sup\left\{\tau \in [0,1] : \sup_{\tau' \leq \tau} \hat{\alpha}_{\text{naive}}(\tau') \leq \alpha\right\}. \tag{2}$$

In turn, the resulting calibrated LPB is given by $\hat{L}(X; \hat{\tau}_{\text{naive}}) = \hat{q}_{\hat{\tau}_{\text{naive}}}(X)$. Notably, this $\hat{L}(X; \hat{\tau}_{\text{naive}})$ is a PAC-type LPB on $\tilde{T}$, and thus also a PAC-type LPB on $T$, since $\tilde{T} \leq T$. While this result is not novel, for completeness, we present Theorem A.1 in the Appendix, which proves its validity.

## 2.2 Utilizing type-I censored data for powerful calibration

Recall that $\hat{\alpha}_{\text{naive}}$ from Equation 1 is an overly conservative miscoverage estimator, particularly when early censoring occurs. This is because $\hat{\alpha}_{\text{naive}}$ counts the events in which either $C_i < \hat{q}_\tau(X_i)$ or $T_i < \hat{q}_\tau(X_i)$ hold. To form a less conservative estimator, Gui et al. (2024) suggest discarding the samples for which $C_i < \hat{q}_\tau(X_i)$, a strategy that is inline with Candès et al. (2023). Under Assumption 1.1, it can be shown that such a selection rule induces a covariate shift. In turn, Gui et al. (2024) introduced the following miscoverage estimator:

$$\hat{\alpha}_{\text{type-I}}(\tau) = \frac{\sum_{i \in \mathcal{I}_{\text{cal}}} \hat{w}_\tau^{\text{type-I}}(X_i) \cdot \mathbb{I}\{\hat{q}_\tau(X_i) \leq C_i\} \cdot \mathbb{I}\{\tilde{T}_i < \hat{q}_\tau(X_i)\}}{\sum_{i \in \mathcal{I}_{\text{cal}}} \hat{w}_\tau^{\text{type-I}}(X_i) \cdot \mathbb{I}\{\hat{q}_\tau(X_i) \leq C_i\}},$$

where the weights $\hat{w}_\tau^{\text{type-I}}(x)$, approximating $1/\mathbb{P}(\hat{q}_\tau(X) \leq C \mid X = x)$, rigorously account for the covariate shift induced by the selection rule $\mathbb{I}\{\hat{q}_\tau(X_i) \leq C_i\}$. Notably, $\hat{\alpha}_{\text{type-I}}(\tau)$ differs from $\hat{\alpha}_{\text{naive}}(\tau)$ from Equation 1 in that the latter uses all the samples for calibration, i.e., the selection rule indicator is always set to 1. Armed with $\hat{\alpha}_{\text{type-I}}(\tau)$, the hyper-parameter $\hat{\tau}_{\text{type-I}}$ can be tuned analogously to Equation 2. Finally, the LPB of a new test point is given by $\hat{L}(X; \hat{\tau}_{\text{type-I}}) = \hat{q}_{\hat{\tau}_{\text{type-I}}}(X)$.

The advantage of the estimator $\hat{\alpha}_{\text{type-I}}(\tau)$ is that it is mostly less conservative compared to $\hat{\alpha}_{\text{naive}}(\tau)$, resulting in more informative LPBs. However, for *general right-censored data*, using $\hat{\alpha}_{\text{type-I}}(\tau)$ is not feasible as the censorship time $C_i$ is not available for some subjects. Our work addresses this knowledge gap, showing how to rigorously utilize the event indicator $e_i = \mathbb{I}\{T_i < C_i\}$ to construct not only valid but also useful LPBs, as detailed in the next section.

## 3 Proposed methods

An intuitive adaptation of the miscoverage estimator proposed by Gui et al. (2024) to *general right-censored data* would be to examine only calibration points for which $e_i = 1$. The challenge with this selection rule is that the distribution shift it induces cannot be formulated as a covariate shift. This is because, even under Assumption 1.1, we expect statistical dependence between $T$ and $e$ conditional on $X$. Furthermore, without careful adjustments, estimating the miscoverage using only uncensored subjects jeopardizes the LPB's validity as the survival times of uncensored subjects are generally lower than the ones of the general population.

Nevertheless, in this section, we show that with a proper weighting scheme, the above set of ideas can be formalized into a valid calibration method. Concretely, we propose two calibration techniques. Our first method, called focused calibration, only uses uncensored subjects for LPB construction. We prove the validity of this method and show that, while it can lead to more informative LPBs than the naive approach, it remains conservative to some extent. This observation drives the proposal of our second approach, which builds upon and further improves the first. Our key idea is to reduce conservativeness by including certain censored examples—in addition to the uncensored subjects—when calibrating the LPB. We term this method fused calibration, as it utilizes both censored and uncensored samples while maintaining the theoretical guarantees.

### 3.1 FOCUSED CALIBRATION

In what follows, we formally introduce our `focused` calibration method, then provide an intuitive explanation for its correctness, and finally characterize the condition under which it leads to LPBs that are more informative than those produced by the `naive` method. Following Gui et al. (2024), we use the LPB function $\hat{L}(X; \tau) = \hat{q}_\tau(X)$, but we tune the hyper-parameter $\tau$ by utilizing the special structure of the general right-censored setting.

In more detail, we calibrate the LPB using samples with $e_i = 1$, i.e., those for which $\tilde{T}_i = T_i$. With this selection rule in place, we define the resulting miscoverage estimator as

$$\hat{\alpha}_{\text{focus}}(\tau) = \frac{\sum_{i \in \mathcal{I}_{\text{cal}}} \hat{w}(X_i) \cdot \mathbb{I}\{e_i = 1\} \cdot \mathbb{I}\{\tilde{T}_i < \hat{q}_\tau(X_i)\}}{\sum_{i \in \mathcal{I}_{\text{cal}}} \hat{w}(X_i) \cdot \mathbb{I}\{e_i = 1\}}, \tag{3}$$

where the weights $\hat{w}(x)$ approximate $1/\mathbb{P}(e = 1 \mid X = x)$. Later, we explain the rationale behind the use of $\hat{w}(x)$ in detail. For now, we remark that: (i) the estimation of $\mathbb{P}(e = 1 \mid X = x)$ can be done by fitting any binary classifier to the training pairs $(X, e)$; and (ii) in Section 3.3, we characterize the effect of the estimation error on the resulting coverage.

Next, the tuned $\hat{\tau}_{\text{focus}}$ is defined as the smallest $\tau$ such that $\hat{\alpha}_{\text{focus}}(\tau)$ passes the user-specified miscoverage level $\alpha$, i.e.

$$\hat{\tau}_{\text{focus}} = \sup \left\{ \tau \in [0, 1] : \sup_{\tau' \leq \tau} \hat{\alpha}_{\text{focus}}(\tau') \leq \alpha \right\}.$$

Finally, the LPB for a new test point is given by $L(X_{\text{test}}; \hat{\tau}_{\text{focus}}) = \hat{q}_{\hat{\tau}_{\text{focus}}}(X_{\text{test}})$. For ease of reference, the algorithm summarizing our `focused` calibration procedure is in Appendix B.3.

Building upon the foundations of Gui et al. (2024), we establish a double robustness result for the validity of our method. This result implies that the LPBs are approximately valid if either (i) the weights $\hat{w}(x)$ are well approximated; or (ii) the estimated quantiles $\hat{q}_\tau(x)$, $\tau \in (0, 1)$ of the conditional distribution $T \mid X$ are accurate. Additionally, if (ii) is satisfied, the LPBs are approximately valid conditional on $X_{\text{test}}$. The theorems formalizing the validity of the `focused` method is postponed to Section 3.3, as this procedure is actually a specific instance of the `fused` calibration method, which is presented in the next section.

Having presented the algorithm, we turn to discuss why the use of the oracle weight $w(x) = 1/\mathbb{P}(e = 1 \mid X = x)$, estimated by $\hat{w}(x)$, conservatively accounts for the distribution shift induced by the selection rule from Equation 3. Our analysis follows Gui et al. (2024), providing an upper bound for the true miscoverage rate $\alpha(\tau)$ obtained by the quantile estimator $\hat{q}_\tau$ for some $\tau \in (0, 1)$:

$$\alpha(\tau) = \mathbb{P}\big(T < \hat{q}_\tau(X)\big) = \mathbb{E}\left[\mathbb{P}(T < \hat{q}_\tau(X) \mid X)\right] \tag{4}$$
$$= \mathbb{E}\left[\mathbb{P}(T < \hat{q}_\tau(X) \mid X) \cdot \mathbb{P}(e = 1 \mid X) \cdot w(X)\right]$$
$$\leq \mathbb{E}\left[\mathbb{P}(T < \hat{q}_\tau(X), e = 1 \mid X) \cdot w(X)\right] \tag{5}$$
$$= \mathbb{E}\left[\mathbb{I}\{T < \hat{q}_\tau(X), e = 1\} \cdot w(X)\right] \tag{6}$$
$$= \mathbb{E}\left[\mathbb{I}\{\tilde{T} < \hat{q}_\tau(X)\} \cdot \mathbb{I}\{e = 1\} \cdot w(X)\right] = \alpha_{\text{focus}}(\tau). \tag{7}$$

Above, steps 4 and 6 hold by the tower property, and the inequality in step 5 is due to Lemma B.1, provided in the Appendix. Importantly, the above derivation shows that, while we do not have direct access to the true $\alpha(\tau)$, we can upper bound it using a weighted average of observed quantities, as revealed by step 7. In turn, the miscoverage upper bound $\alpha_{\text{focus}}(\tau)$ provides the basis for constructing its empirical estimator $\hat{\alpha}_{\text{focus}}(\tau)$ from Equation 3, with the denominator serving to normalize the weighted average.

While being conservative, our experiments show that the LPBs derived from the `focused` method tend to be more informative than those produced by the `naive` approach from Section 2.1. To better understand when this is the case, we now present a condition under which $\alpha_{\text{focus}}(\tau)$ is less conservative than $\alpha_{\text{naive}}(\tau)$.

**Proposition 3.1.** *Under Assumption 1.1, the relation $\alpha_{\text{focus}}(\tau) < \alpha_{\text{naive}}(\tau)$ holds when*

$$\mathbb{P}(\tilde{T} < \hat{q}_\tau(X) \mid X = x, e = 1) < \mathbb{P}(\tilde{T} < \hat{q}_\tau(X) \mid X = x, e = 0) \quad \forall x \in \mathcal{X}. \tag{8}$$

The proof is given in Appendix B.2. To highlight the practical implications of the above proposition, we recall the definition of $\tilde{T} = \min(T, C)$ and $e$, and re-write Condition 8 as follows:

$$\mathbb{P}(T < \hat{q}_\tau(X) \mid X = x, e = 1) < \mathbb{P}(C < \hat{q}_\tau(X) \mid X = x, e = 0) \quad \forall x \in \mathcal{X}.$$

As such, our `focused` method is anticipated to outperform the `naive` approach when the probability of early survival time for subjects with $e = 1$ is smaller than the probability of early censorship time for subjects with $e = 0$. Such a phenomenon can occur in medical trials, where early censoring is prevalent due to factors such as non-compliance (Zhou et al., 2020), withdrawal of consent (Wilson et al., 2021), and loss to follow-up (Fontana et al., 2018; Gill et al., 2018; Monfared et al., 2021).

With that said, Proposition 3.1 also reveals that the `naive` approach from Section 2.1 may produce tighter LPBs than the `focused` method. Specifically, this occurs for samples that do not satisfy the inequality in Equation 8. This insight naturally brings the idea of detecting such samples and leveraging them to mitigate the conservativeness of $\hat{\alpha}_{\text{focus}}(\tau)$. Indeed, this is the key principle behind our `fused` calibration method, described hereafter.

## 3.2 FUSED CALIBRATION

Following the above discussion, we show how to fuse the `naive` calibration approach with the `focused` calibration method to enhance statistical efficiency, making better use of available data. In addition to using all the uncensored points for calibration, we aim to selectively include in the calibration set the censored subjects that violate the inequality in Equation 8. The oracle selection criterion is thus formally stated as

$$s_\tau(x) = \begin{cases} 1, & \text{if } \mathbb{P}(\tilde{T} < \hat{q}_\tau(X) \mid X = x, e = 0) < \mathbb{P}(\tilde{T} < \hat{q}_\tau(X) \mid X = x, e = 1), \\ 0, & \text{otherwise.} \end{cases}$$

Observe that we cannot compute $s_\tau(x)$ in practice as we do not have access to the conditional distribution $\tilde{T} \mid X, e$. However, this indicator $s_\tau(x)$ can be estimated using a single classifier, as follows. First, train a binary classifier with $(X_i, e_i)$ as input, predicting the label $\mathbb{I}\{\tilde{T}_i < \hat{q}_\tau(X_i)\}$. Then, use this classifier to compare the estimated probabilities of the label given $(X = x, e = 0)$ and $(X = x, e = 1)$. We refer to the fitted estimator of $s_\tau(x)$ as $\hat{s}_\tau(x)$. Crucially, inaccurate estimation of $\hat{s}_\tau$ does not impact the validity of the calibration procedure, however it can affect the gain in statistical efficiency.

With $\hat{s}_\tau(x)$ in place, we formulate the fused selection rule, which uses the calibration point $(X_i, \tilde{T}_i)$ for miscoverage estimation if either $e_i = 1$ or $\hat{s}_\tau(X_i) = 1$:

$$\zeta_\tau(X_i, e_i) = \begin{cases} 1, & \text{if } \hat{s}_\tau(X_i) = 1, \\ e_i, & \text{otherwise.} \end{cases} \tag{9}$$

Next, the fused miscoverage estimator is formulated as follows:

$$\hat{\alpha}_{\text{fused}}(\tau) = \frac{\sum_{i \in \mathcal{I}_{\text{cal}}} \hat{w}_\tau(X_i) \cdot \mathbb{I}\{\zeta_\tau(X_i, e_i) = 1\} \cdot \mathbb{I}\{\tilde{T}_i < \hat{q}_\tau(X_i)\}}{\sum_{i \in \mathcal{I}_{\text{cal}}} \hat{w}_\tau(X_i) \cdot \mathbb{I}\{\zeta_\tau(X_i, e_i) = 1\}},$$

where the weights $\hat{w}_\tau(x)$ approximate $w_\tau(x) = 1/\mathbb{P}(\zeta_\tau(x, e) = 1 \mid X = x)$. We note that since $\hat{s}_\tau(X_i)$ is deterministic given $X_i$, we have that

$$w_\tau(X_i) = \begin{cases} 1, & \text{if } \hat{s}_\tau(X_i) = 1, \\ 1/\mathbb{P}(e_i = 1 \mid X_i), & \text{otherwise.} \end{cases}$$

Continuing with the same rationale as in `focused` calibration, we define $\hat{\tau}_{\text{fused}}$ as

$$\hat{\tau}_{\text{fused}} = \sup \left\{ \tau \in [0, 1] : \sup_{\tau' \leq \tau} \hat{\alpha}_{\text{fused}}(\tau') \leq \alpha \right\},$$

and set the LPB for a new test point as $\hat{L}(X_{\text{test}}; \hat{\tau}_{\text{fused}}) = \hat{q}_{\hat{\tau}_{\text{fused}}}(X_{\text{test}})$.

For ease of reference, the `fused` calibration method is presented in Algorithm 3. Notably, we use training data (independent from the calibration points) to fit the estimated quantile functions $\hat{q}_\tau(x)$,

the classifiers $\hat{s}_\tau(x)$, and the weights $\hat{w}_\tau(x)$; these models serve as inputs to the `fused` calibration algorithm.

We remark that Algorithm 3 reduces to the `focused` approach with the choice of $\hat{s}_\tau(x) := 0$ for all $x$, and to the `naive` approach when $\hat{s}_\tau(x) := 1$ for all $x$. For this reason, we argue that the `fused` method can be viewed as an interpolation between two valid calibration strategies, with the optimal interpolation hyperparameter being estimated by $\hat{s}_\tau(x)$. Indeed, our theory shows that the `fused` method is valid regardless of how accurately the model $\hat{s}_\tau(x)$ estimates $s_\tau(x)$. This is because any selection rule, determined by the model $\hat{s}_\tau(x)$, yields a conservative estimate of the miscoverage; see Appendix D.1 as well as the formal results given in the next section. Although validity is maintained, we may lose power when using an inaccurate $\hat{s}_\tau(x)$. To see this, imagine an extreme case where for any $\tau$ the `focused` calibration approach is less conservative than the `naive` approach, i.e., $s_\tau(x) = 0$ for all $x$, and yet our estimate $\hat{s}_\tau(x)$ always predicts the (incorrect) value 1. In such a case, the `fused` method reduces to the `naive` approach, resulting in power loss, but maintaining validity.

### 3.3 THEORETICAL GUARANTEES

Our theoretical framework builds on the work by Gui et al. (2024) and adapts their double robustness result to our proposed methods for general right-censored data. Concretely, our first validity result states that if the weights are well-estimated, i.e., $\hat{w}_\tau(x) \approx w_\tau(x)$, then the resulting coverage is approximately higher than $1 - \alpha$ conditional on the training data $\mathcal{D}_{\text{tr}} = \{(X_i, \tilde{T}_i, e_i)\}_{i \in \mathcal{I}_{\text{tr}}}$ and the holdout calibration data $\mathcal{D}_{\text{cal}} = \{(X_i, \tilde{T}_i, e_i)\}_{i \in \mathcal{I}_{\text{cal}}}$, where $\mathcal{D} = \mathcal{D}_{\text{tr}} \cup \mathcal{D}_{\text{cal}}$. In particular, this guarantee holds even when the estimated conditional quantiles $\hat{q}_\tau$ are inaccurate. Further, following Theorem 3.1 below, we can see that the guaranteed coverage rate gets closer to the desired level as (i) $\hat{w}_\tau$ gets closer to $w_\tau$; and (ii) the size of the calibration set $|\mathcal{I}_{\text{cal}}|$ increases.

**Theorem 3.1** (Approximate calibration with accurate weights). *Fix any $\alpha, \delta \in (0, 1)$. Given that $\hat{q}_\tau(x)$ is non-decreasing and continuous in $\tau$, and that there exists some constant $\hat{\gamma}_\tau > 0$ such that $\hat{w}_\tau(x) \leq \hat{\gamma}_\tau$ for $P_X$-almost all $x$. Under Assumption 1.1, with probability at least $1 - \delta$ over the draw of $\mathcal{D}$, the LPB $\hat{L}$ produced by either Algorithm 2 or by Algorithm 3 satisfies*

$$\mathbb{P}\left[T \geq \hat{L}(X) | \mathcal{D}\right] \geq 1 - \alpha - \Delta_w,$$

*where*

$$\Delta_w := \sup_{\tau \in [0,1]} \left( \mathbb{E}\left[ \left| \frac{\hat{w}_\tau(X_i)}{w_\tau(X_i)\pi_\tau} - 1 \right| \Big| \mathcal{D}_{\text{tr}} \right] + \sqrt{\frac{1 + \frac{\hat{\gamma}_\tau^2}{\pi_\tau^2} + \max\left(1, \frac{\hat{\gamma}_\tau^2}{\pi_\tau^2} - 1\right)^2}{0.4\,|\mathcal{I}_{\text{cal}}|} \ln\frac{1}{\delta}} \right),$$

*and $\pi_\tau := \mathbb{E}\left[\frac{\hat{w}_\tau(X)}{w_\tau(X)} | \mathcal{D}_{\text{tr}}\right]$.*

The proof is given in Appendix D.2.

The next result states a stronger, approximated conditional coverage guarantee that holds even when the weights are inaccurate. In essence, this result holds under a stronger assumption that the conditional quantiles are well-estimated, i.e., $\hat{q}_\tau(x) \approx q_\tau(x)$. This is stated formally in assumption (b). Before formalizing this intuition, we define the smallest estimated quantile level to bound $T$ with probability at least $\beta$, as

$$\tau(\beta) = \sup\left\{\tau' \in [0, 1] : \mathbb{P}\left(T < \hat{q}_{\tau'}(X) \mid \mathcal{D}_{\text{tr}}\right) \leq \beta\right\}.$$

**Theorem 3.2** (Approximate calibration with accurate conditional quantiles). *Assume the same conditions as Theorem 3.1 and assume further that the distribution of $T \mid X$ is continuous and its density is upper bounded by a constant $B > 0$, and that there exists a constant $r > 0$ such that*

*(a) For $P_X$-almost all $x$: $\sup_{\xi \in [\tau(\alpha), \tau(\alpha+r)+\psi]} w_\xi(x) \leq \gamma$ and $\sup_{\xi \in [\tau(\alpha), \tau(\alpha+r)+\psi]} \hat{w}_\xi(x) \leq \hat{\gamma}$ for some constants $\gamma, \hat{\gamma}, \psi \geq 0$*

*(b) $\sup_{\beta \in [\alpha, \alpha+r]} \sup_{x \in \mathcal{X}} \left\{\max(B, 1)|\hat{q}_{\tau(\beta)}(x) - q_\beta(x)| + \hat{\gamma}\gamma\sqrt{\frac{\log(1/\delta)}{0.4|\mathcal{I}_{\text{cal}}|}}\right\} \leq r$, where $q_\beta(x)$ is the $\beta$-quantile of the distribution of $T \mid X = x$.*

*Then, under Assumption 1.1, with probability at least $1 - \delta$ over the draw of $\mathcal{D}$, the LPB $\hat{L}$ produced by either Algorithm 2 or by Algorithm 3 satisfies that for $P_X$-almost all $x$,*

$$\mathbb{P}\left[T \geq \hat{L}(X)|\mathcal{D}, X = x\right] \geq 1 - \alpha - \Delta_q,$$

*where $\Delta_q = \sup_{\beta \in [\alpha, \alpha+r]} \sup_{x \in \mathcal{X}} \left\{2B \cdot \left|\hat{q}_{\tau(\beta)}(x) - q_\beta(x)\right|\right\} + \hat{\gamma}\gamma\sqrt{\frac{1}{0.4|\mathcal{I}_{cal}|}\log\frac{1}{\delta}}.$*

The proof is deferred to Appendix D.3.

## 4 EXPERIMENTS

### 4.1 SIMULATION STUDIES

We follow the experimental protocol outlined by Candès et al. (2023) and Gui et al. (2024) to evaluate the performance of our methods across various simulated scenarios. Notably, since real-world data lack the true survival time $T$ for all test points, synthetic data experiments are important to verify the correctness of our theory.

**Base predictive models** In all experiments, we utilize the *DeepSurv* method (Katzman et al., 2018), as the base predictor to estimate the conditional quantiles $\hat{q}_\tau$. We also fit Random Forest classifiers to estimate the weights $\hat{w}_\tau$ and indicator $\hat{s}_\tau$ for our calibration methods. For more details, see Appendix F.

**Data** We test our calibration methods on six simulated settings designed to mimic real-world clinical trials, as in Candès et al. (2023); Gui et al. (2024). Refer to Appendix E.1 for a details.

**Methods and performance metrics** To evaluate our approach, we compare our method to the base model, applied without calibration, and to the CSD calibration method proposed by Qi et al. (2024). In all simulation studies, we set the desired coverage level $1 - \alpha$ to 90% and evaluate both the coverage rate and the average LPB over the test set.

**Results** The performance metrics for the synthetic experiments are presented in Figure 1. Following the top panel in that figure, we can see that the `naive`, `focused`, and `fused` calibration methods achieve valid coverage in all six settings. This stands in contrast to both the uncalibrated base model and the CSD methods which produce LPBs that do not obtain the desired coverage level in settings 1, 3, and 4. Notably, our `fused` method tends to be less conservative than the `focused` and `naive` approaches, highlighting the advantage of the `fused` approach. Settings 2 and 6 further stress the advantage of the `fused` method, where one can see that (i) the `focused` approach is less conservative than the `naive` approach, suggesting a violation of Equation 8; and (ii) the `fused` method is more powerful than the `focused` and `naive` methods.

Turning to LPB comparisons, the bottom panel of Figure 1 illustrates that the `fused` method generates the most informative lower bounds out of all the methods with finite sample guarantees (higher is better). This statistical gain is in line with the tighter coverage rate of the `fused` method. The uncalibrated baseline and CSD calibration methods produce the highest LPBs, but these LPBs are often invalid.

**Additional synthetic experiments**, demonstrating the robustness of our method to varying censorship rates, are given in Appendix G. Experiments demonstrating the effect of the sample size on the performance of our methods are presented in Appendix H. An ablation study, showing the robustness of our methods to changes in key hyperparameters of the underlying predictive models, is detailed in Appendix I.

### 4.2 APPLICATION TO REAL DATA

We demonstrate the practical utility of our methods by applying them to six real-world datasets: The Northern Alberta Cancer Dataset (NACD) (Haider et al., 2020), Rotterdam & German Breast Cancer Study Group (GBSG), Molecular Taxonomy of Breast Cancer International Consortium

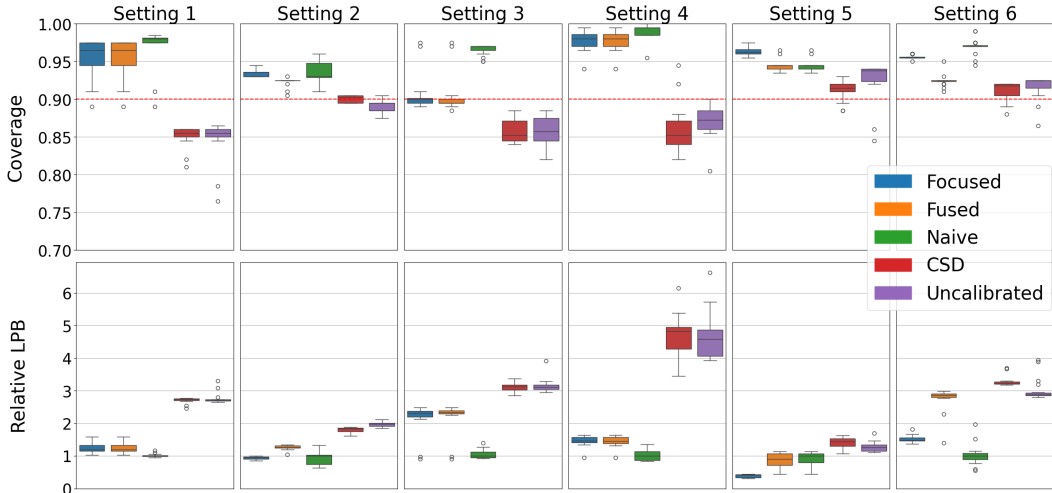

Figure 1: Performance of the different calibration methods for each of the six simulated settings. **Top**: empirical coverage rate, with a red dashed line indicating the nominal 90% level. **Bottom**: relative LPB, defined as the LPB obtained by each method divided by the `naive` method's median LPB. A higher relative LPB is better. The reported performance metrics are evaluated on 50 independent trials, each consisting of newly sampled train, validation, calibration, and test sets of sizes 600, 200, 1000, and 200, respectively. The notably large proportion of calibration samples is intentionally chosen to ensure that the CSD method has sufficient data, as advised by Qi et al. (2024).

(METABRIC), Study to Understand Prognoses Preferences Outcomes and Risks of Treatment (SUPPORT) (Kvamme et al., 2019; Katzman et al., 2018), a user churn dataset (Fotso et al., 2019–present), as well as The Cancer Genome Atlas Breast Invasive Carcinoma (TCGA-BRCA) multimodal dataset collection (Tomczak et al., 2015). The datasets' description are detailed in Appendix E and the TCGA-BRCA dataset's acquisition and preprocessing are detailed in Appendix F. Censorship rates for all datasets are detailed in Table 3.

In contrast with the synthetic experiments, here it is not possible to rigorously verify that the desired coverage level is achieved in practice—we do not have access to ground truth survival times. Yet, our theory indicates that the `naive` method should attain a conservative coverage level under minimal assumptions. Further, under well-approximated estimation of the weights or quantiles, the proposed `focused` and `fused` methods should produce less conservative and approximately valid LPBs.

To evaluate how the various methods perform in practice, we present in Figure 2 their corresponding average LPB values. Among the methods that provide finite-sample guarantees, the `fused` approach achieves LPB values comparable to the stronger-performing method between the `naive` and `focused`. Furthermore, the `fused` method outperforms both the `naive` and `focused` methods, albeit to a limited extent, on the METABRIC dataset; this is reminiscent of settings 2 and 6 in the synthetic experiments. Finally, we note that while there are datasets for which the uncalibrated base model and the CSD approach yield higher LPB values, our synthetic experiments highlighted that there are cases where these methods fail to achieve the desired coverage in practice.

## 5 DISCUSSION

We introduced a flexible uncertainty quantification framework with finite-sample LPB coverage guarantees for general right-censored survival analysis data. The validity of the proposed calibration methods relies on the conditionally independent censoring and i.i.d. assumptions, which may not hold in all real-world cases. It is therefore of great importance to study the effect of violations of these assumptions both to understand the practical limitations of our methods and to further enhance their robustness. In particular, the work by Oliveira et al. (2024) may serve as a valuable starting

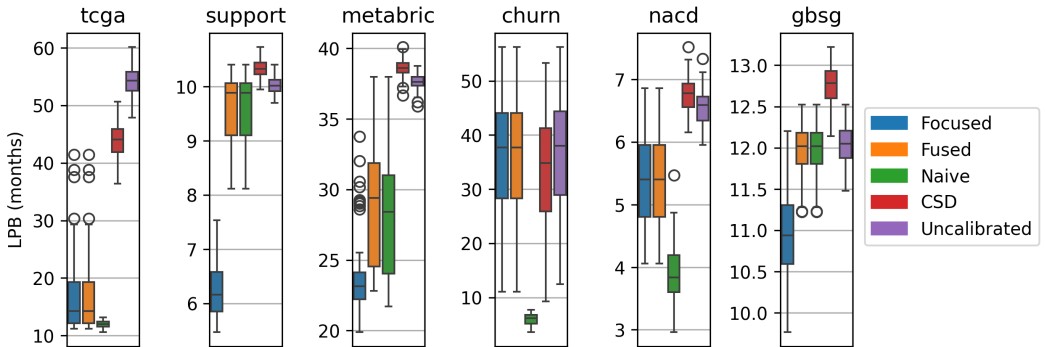

Figure 2: Real-world data experiments: comparison of the average LPB obtained by different calibration methods. For each dataset, base models are trained once using randomly selected training and validation sets, comprising 60% and 15% of the dataset, respectively. The remaining dataset (25%) forms the calibration (10%) and test (15%) sets. LPBs are calculated over 100 random calibration/test splits.

point to move beyond the i.i.d. assumption; and the work by Feldman & Romano (2024) can be further utilized to build robustness to imperfect training data.

We should also emphasize that our methods may fail to generate valid LPBs in cases where the weights $\hat{w}_\tau(x)$ are poorly estimated. Stressing this point, the estimation of $\hat{w}_\tau(x)$ can be particularly challenging when the censoring rate is high, resulting in imbalanced training data. A related issue is handling extreme $\hat{w}_\tau(x)$, where one possible remedy is bound the estimate of $\mathbb{P}(e = 1 \mid X)$ in a manner similar to Gui et al. (2024). It would be of great interest to offer more principled solutions to the above challenges, possibly by following the ideas of Li et al. (2018); Cheng et al. (2022); Elze et al. (2017).

# 6 ACKNOWLEDGMENTS

This research was partially supported by the Israel Science Foundation (ISF grant 3864/21), the Israel Cancer Research Fund (ICRF grant 1281495), and the Zimin Insititute for AI solutions in Healthcare. Y. R., S. F., and H. D. were supported by the Israel Science Foundation (ISF grant 729/21). Y. R., S. F., and H. D. also thank the European Union (ERC, SafetyBounds, 101163414) for providing additional support. Views and opinions expressed are however those of the authors only and do not necessarily reflect those of the European Union or the European Research Council Executive Agency. Neither the European Union nor the granting authority can be held responsible for them. Y. R. thanks the Career Advancement Fellowship, Technion, for providing additional research support.

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

## A    Naive method's algorithm and proof of validity

A formal description of the algorithm for the `naive` calibration method is given by Algorithm 1.

---

**Algorithm 1** Conformalized survival analysis for general right-censored data: `naive` calibration

---

**Input:** desired miscoverage level $\alpha$; calibration data $\mathcal{D}_{\text{cal}} = \{(X_i, \tilde{T}_i, e_i)\}_{i \in \mathcal{I}_{\text{cal}}}$; estimated quantiles of $T \mid X$, $\{\hat{q}_\tau(\cdot)\}_{\tau \in [0,1]}$.

**Procedure:**

 1: **for** $\tau$ in a grid over $[0, 1]$ **do**
 2:     Compute the fused miscoverage estimator:

$$\hat{\alpha}_{\text{naive}}(\tau) = \frac{1}{|\mathcal{I}_{\text{cal}}|} \sum_{i \in \mathcal{I}_{\text{cal}}} \mathbb{I}\{\tilde{T}_i < \hat{q}_\tau(X_i)\}$$

 3: **end for**
 4: Calibrate the hyperparameter $\tau$:

$$\hat{\tau}_{\text{naive}} = \sup\left\{\tau \in [0, 1] : \sup_{\tau' \leq \tau} \hat{\alpha}_{\text{naive}}(\tau') \leq \alpha\right\}$$

 5: **Return:** The calibrated LPB: $\hat{L}(\cdot) := \hat{L}(\cdot; \hat{\tau}_{\text{naive}}) = \hat{q}_{\hat{\tau}_{\text{naive}}}(\cdot)$

---

**Theorem A.1** (PAC-type validity of the naive LPB). *Fix any $\alpha, \delta \in (0, 1)$. Assume that $\hat{q}_\tau(x)$ is non-decreasing in $\tau$. With probability at least $1 - \delta$ over the draw of $\mathcal{D}$, the LPB produced by the* `naive` *calibration method from satisfies*

$$\mathbb{P}\left[T \geq \hat{L}(X; \hat{\tau}_{\text{naive}})|\mathcal{D}\right] \geq 1 - \alpha - \Delta_n,$$

*where $\Delta_n := \sqrt{\frac{1}{|\mathcal{I}_{\text{cal}}|} \ln \frac{1}{\delta}}$.*

*Proof.* For the proof we denote the smallest estimated quantile to bound $T$ with probability at least $\beta$, conditional on the training data, as

$$\tau(\beta) = \sup \{\tau' \in [0, 1] : \mathbb{P}(T < \hat{q}_{\tau'}(X) \mid \mathcal{D}_{\text{tr}}) \leq \beta\}.$$

As in the proof for theorem 3 in the work by Gui et al. (2024), proving that $1 - \delta \le \mathbb{P}[\hat{\tau}_{\text{naive}} \le \tau(\alpha + \Delta_n) \mid \mathcal{D}_{\text{tr}}]$ is sufficient for proving the theorem. Analyzing the `naive` miscoverage estimator, we get that

$$\mathbb{P}[\hat{\alpha}_{\text{naive}}(\tau(\alpha + \Delta_n) + \epsilon) \le \alpha \mid \mathcal{D}_{\text{tr}}]$$

$$= \mathbb{P}\left[\sum_{i \in \mathcal{I}_{\text{cal}}} \mathbb{I}\left\{\tilde{T}_i < \hat{q}_{\tau(\alpha + \Delta_n) + \epsilon}(X_i)\right\} \le |\mathcal{I}_{\text{cal}}| \alpha \,\middle|\, \mathcal{D}_{\text{tr}}\right]$$

$$= \mathbb{P}\left[\sum_{i \in \mathcal{I}_{\text{cal}}} \left(\mathbb{I}\left\{\tilde{T}_i < \hat{q}_{\tau(\alpha + \Delta_n) + \epsilon}(X_i)\right\} - \alpha\right) \le 0 \,\middle|\, \mathcal{D}_{\text{tr}}\right]$$

$$\le \mathbb{E}\left[t \exp\left(\sum_{i \in \mathcal{I}_{\text{cal}}} \left(\alpha - \mathbb{I}\left\{\tilde{T}_i < \hat{q}_{\tau(\alpha + \Delta_n) + \epsilon}(X_i)\right\}\right)\right) \,\middle|\, \mathcal{D}_{\text{tr}}\right] \qquad (10)$$

where the first transition is by the definition of $\hat{\alpha}_{\text{naive}}$, and the last by the Markov inequality. Further conditioning on $(X_i)_{i \in \mathcal{I}_{\text{cal}}}$, since $p_{\tau(\alpha + \Delta_n) + \epsilon}(X_i) - \mathbb{I}\left\{\tilde{T}_i < \hat{q}_{\tau(\alpha + \Delta_n) + \epsilon}(X_i)\right\}$ is $\frac{1}{4}$-sub-gaussian, we can use Hoeffding's lemma we get that

$$\mathbb{E}\left[t \exp\left(\sum_{i \in \mathcal{I}_{\text{cal}}} \left(p_{\tau(\alpha + \Delta_n) + \epsilon}(X_i) - \mathbb{I}\left\{\tilde{T}_i < \hat{q}_{\tau(\alpha + \Delta_n) + \epsilon}(X_i)\right\}\right)\right) \,\middle|\, (X_i)_{i \in \mathcal{I}_{\text{cal}}}, \mathcal{D}_{\text{tr}}\right]$$

$$\le \exp\left(t \sum_{i \in \mathcal{I}_{\text{cal}}} \mathbb{E}\left[p_{\tau(\alpha + \Delta_n) + \epsilon}(X_i) - \mathbb{I}\left\{\tilde{T}_i < \hat{q}_{\tau(\alpha + \Delta_n) + \epsilon}(X_i)\right\} \,\middle|\, (X_i)_{i \in \mathcal{I}_{\text{cal}}}, \mathcal{D}_{\text{tr}}\right] + \frac{t^2}{8} \sum_{i \in \mathcal{I}_{\text{cal}}} 1^2\right)$$

$$= \exp\left(t \sum_{i \in \mathcal{I}_{\text{cal}}} \left(p_{\tau(\alpha + \Delta_n) + \epsilon}(X_i) - \mathbb{P}\left\{\tilde{T}_i < \hat{q}_{\tau(\alpha + \Delta_n) + \epsilon}(X_i)\right\} \,\middle|\, (X_i)_{i \in \mathcal{I}_{\text{cal}}}, \mathcal{D}_{\text{tr}}\right] + \frac{t^2 |\mathcal{I}_{\text{cal}}|}{8}\right).$$

Since $\tilde{T}_i \le T_i$ by definition, we have that

$$p_{\tau(\alpha + \Delta_n) + \epsilon}(X_i) - \mathbb{P}\left\{\tilde{T}_i < \hat{q}_{\tau(\alpha + \Delta_n) + \epsilon}(X_i)\right\} \le 0,$$

and so by the law of total expectation, we get that

$$\mathbb{E}\left[t \exp\left(\sum_{i \in \mathcal{I}_{\text{cal}}} \left(p_{\tau(\alpha + \Delta_n) + \epsilon}(X_i) - \mathbb{I}\left\{\tilde{T}_i < \hat{q}_{\tau(\alpha + \Delta_n) + \epsilon}(X_i)\right\}\right)\right) \,\middle|\, \mathcal{D}_{\text{tr}}\right]$$

$$\le \exp\left(\frac{|\mathcal{I}_{\text{cal}}| t^2}{8}\right).$$

Combining with 10, we get that

$$\mathbb{P}[\hat{\alpha}_{\text{naive}}(\tau(\alpha + \Delta_n) + \epsilon) \le \alpha \mid \mathcal{D}_{\text{tr}}]$$

$$\le \exp\left(\frac{|\mathcal{I}_{\text{cal}}| t^2}{8}\right) \mathbb{E}\left[t \exp\left(\sum_{i \in \mathcal{I}_{\text{cal}}} \left(\alpha - p_{\tau(\alpha + \Delta_n) + \epsilon}(X_i)\right)\right) \,\middle|\, \mathcal{D}_{\text{tr}}\right].$$

By the definition of $\tau(\alpha + \Delta_n)$, we have that

$$\mathbb{P}\left[T < \hat{q}_{\tau(\alpha + \Delta_n) + \epsilon}(X) \mid \mathcal{D}_{\text{tr}}\right] \le \alpha + \Delta_n,$$

and so,

$$\mathbb{P}\left[\hat{\alpha}_{\text{naive}}\left(\tau\left(\alpha+\Delta_n\right)+\epsilon\right)\le\alpha\mid\mathcal{D}_{\text{tr}}\right]$$

$$\le\exp\left(\frac{|\mathcal{I}_{\text{cal}}|\,t^2}{8}\right)\mathbb{E}\left[t\exp\left(\sum_{i\in\mathcal{I}_{\text{cal}}}\left(\mathbb{P}\left[T<\hat{q}_{\tau(\alpha+\Delta_n)+\epsilon}\left(X_i\right)\mid\mathcal{D}_{\text{tr}}\right]\right.\right.\right.$$

$$\left.\left.\left.-\Delta_n-p_{\tau(\alpha+\Delta_n)+\epsilon}\left(X_i\right)\right)\right)\bigg|\mathcal{D}_{\text{tr}}\right]$$

$$\le\exp\left(\frac{|\mathcal{I}_{\text{cal}}|\,t^2}{8}-t\Delta_n\,|\mathcal{I}_{\text{cal}}|\right),$$

where the last inequality follows from the $\frac{1}{4}$-sub-gaussianity of

$$p_{\tau(\alpha+\Delta_n)+\epsilon}\left(X_i\right)-\mathbb{P}\left[T<\hat{q}_{\tau(\alpha+\Delta_n)+\epsilon}\left(X_i\right)\mid\mathcal{D}_{\text{tr}}\right].$$

Taking

$$t=4\Delta_n,$$

we get that

$$\mathbb{P}\left[\hat{\alpha}_{\text{naive}}\left(\tau\left(\alpha+\Delta_n\right)+\epsilon\right)\le\alpha\mid\mathcal{D}_{\text{tr}}\right]\le\exp\left(-2\,|\mathcal{I}_{\text{cal}}|\,(\Delta_n)^2\right).$$

Since $\left(-2\,|\mathcal{I}_{\text{cal}}|\,(\Delta_n)^2\right)\le-\dfrac{|\mathcal{I}_{\text{cal}}|\cdot\left(\mathbb{E}\left[\left|\frac{\hat{w}(X)}{w(X)}-\pi\right|\middle|\mathcal{D}_{\text{tr}}\right]-\pi\Delta_n\right)^2}{\pi^2+\hat{\gamma}^2+\tilde{\gamma}^2}$, we get that

$$\mathbb{P}\left[\hat{\alpha}_{\text{naive}}\left(\tau\left(\alpha+\Delta_n\right)+\epsilon\right)\le\alpha\mid\mathcal{D}_{\text{tr}}\right]\le\delta,$$

where the last inequality follows from the definition of $\Delta_n$. Taking the limit as $\epsilon\to0$, we get that

$$1-\delta\le\mathbb{P}\left[\hat{\alpha}_{\text{naive}}\left(\tau\left(\alpha+\Delta_n\right)\right)>\alpha\mid\mathcal{D}_{\text{tr}}\right]\le\mathbb{P}\left[\hat{\tau}_{\text{naive}}<\tau\left(\alpha+\Delta_n\right)\mid\mathcal{D}_{\text{tr}}\right],$$

as needed. $\qquad\square$

# B  FURTHER DETAILS REGARDING THE FOCUSED CALIBRATION METHOD

## B.1  FOCUSED CALIBRATION LEMMA

**Lemma B.1.** *Suppose that $x\in\mathcal{X},q:\mathcal{X}\to\mathbb{R}^+$. If $\mathbb{P}(T<q(X)\mid X=x)>0$ and under Assumtion 1.1,*

$$\mathbb{P}\left(e=1,T<q(X)\,|\,X=x\right)\ge\mathbb{P}\left(e=1\,|\,X=x\right)\mathbb{P}\left(T<q(X)\,|\,X=x\right),$$

*where the probability is taken over $P_{(X,C,T)}$.*

*Proof.* Suppose $x\in\mathcal{X}$. All derivations in this proof are conditional on $X=x$, and so we denote $\mathbb{P}_x(\cdot)=\mathbb{P}(\cdot\mid X=x)$ and $q=q(X)$.

We begin by developing $\mathbb{P}_x(T<C\mid T<q)$:

$$\mathbb{P}_x(T<C\mid T<q)=\mathbb{P}_x(T<C\mid T<q,C\ge q)\mathbb{P}_x(C\ge q)+\mathbb{P}_x(T<C\mid T<q,C<q)\mathbb{P}_x(C<q)$$

$$=1\cdot\mathbb{P}_x(C\ge q)+\frac{\mathbb{P}_x(T<C,T<q,C<q)}{\mathbb{P}_x(T<q,C<q)}\mathbb{P}_x(C<q)$$

$$=\mathbb{P}_x(C\ge q)+\frac{\mathbb{P}_x(T<C,T<q,C<q)}{\mathbb{P}_x(T<q)\cdot\mathbb{P}_x(C<q)}\mathbb{P}_x(C<q)$$

$$=\mathbb{P}_x(C\ge q)+\frac{\mathbb{P}_x(T<C,T<q,C<q)}{\mathbb{P}_x(T<q)}$$

$$=\mathbb{P}_x(C\ge q)+\frac{\mathbb{P}_x(T<C,C<q)}{\mathbb{P}_x(T<q)},$$

where the first transition is by the law of total probability and the third is by Assumption 1.1.

We now develop $\mathbb{P}_x(T < C)$:

$$\mathbb{P}_x(T < C) = \mathbb{P}_x(T < C, C < q) + \mathbb{P}_x(T < C, C \geq q)$$

By subtraction the two quantities, we get

$$\mathbb{P}_x(T < C \mid T < q) - \mathbb{P}_x(T < C)$$

$$= \mathbb{P}_x(C \geq q) + \frac{\mathbb{P}_x(T < C, C < q)}{\mathbb{P}_x(T < q)} - (\mathbb{P}_x(T < C, C < q) + \mathbb{P}_x(T < C, C \geq q))$$

$$= (\mathbb{P}_x(C \geq q) - \mathbb{P}_x(T < C, C \geq q)) + (\frac{\mathbb{P}_x(T < C, C < q)}{\mathbb{P}_x(T < q)} - \mathbb{P}_x(T < C, C < q))$$

$$= \mathbb{P}_x(T \geq C, C \geq q) + \left( \frac{1}{\mathbb{P}_x(T < q)} - 1 \right) \mathbb{P}_x(T < C, C < q) \geq 0,$$

and so, multiplying by $\mathbb{P}_x(T < q)$ we get that

$$\mathbb{P}_x(T < C, T < q) - \mathbb{P}_x(T < C)\mathbb{P}_x(T < q) \geq 0.$$

That is,

$$\mathbb{P}_x(T < C, T < q) \geq \mathbb{P}_x(T < C)\mathbb{P}_x(T < q).$$

$\square$

## B.2 FOCUSED CALIBRATION INFORMATIVENESS CONDITION

In this section, we prove Proposition 3.1.

*Proof.* By looking at 5, we have that

$$\alpha_{\text{focus}}(\tau) = \mathbb{E}\left[\mathbb{P}(T < \hat{q}_\tau(X), e = 1 \mid X) \cdot w(X)\right] = \mathbb{E}\left[\mathbb{P}(\tilde{T} < \hat{q}_\tau(X) \mid X, e = 1)\right].$$

Analogously, we can re-write $\alpha_{\text{naive}}(\tau)$ as

$$\alpha_{\text{naive}}(\tau) = \mathbb{P}(\tilde{T} < \hat{q}_\tau(X)) = \mathbb{E}\left[\mathbb{P}(\tilde{T} < \hat{q}_\tau(X) \mid X)\right].$$

Therefore, $\alpha_{\text{focus}}(\tau) < \alpha_{\text{naive}}(\tau)$ when

$$\mathbb{P}(\tilde{T} < \hat{q}_\tau(X) \mid X = x, e = 1) < \mathbb{P}(\tilde{T} < \hat{q}_\tau(X) \mid X = x) \quad \forall x \in \mathcal{X}.$$

Which, by the law of total probability, can be stated as

$$\mathbb{P}(\tilde{T} < \hat{q}_\tau(X) \mid X = x, e = 1) < \mathbb{P}(\tilde{T} < \hat{q}_\tau(X) \mid X = x, e = 0) \quad \forall x \in \mathcal{X}.$$

$\square$

## B.3 FORMAL DESCRIPTION OF THE FOCUSED CALIBRATION ALGORITHM

A formal description of the focused calibration algorithm is given in Algorithm 2.

Note that the theorems for the validity of this algorithm in Section 3.3 are valid under the definition $\forall \tau \in [0, 1], w_\tau := w, \hat{w}_\tau := \hat{w}$.

## C PSEUDO-CODE DESCRIPTION OF THE FUSED CALIBRATION METHOD

A formal description of the fused calibration method is presented in Algorithm 3.

---

**Algorithm 2** Conformalized survival analysis for general right-censored data: `focused` calibration

---

**Input:** desired miscoverage level $\alpha$; calibration data $\mathcal{D}_{\text{cal}} = \{(X_i, \tilde{T}_i, e_i)\}_{i \in \mathcal{I}_{\text{cal}}}$; estimated quantiles of $T \mid X$, $\{\hat{q}_\tau(\cdot)\}_{\tau \in [0,1]}$; weights $\hat{w}(x)$ approximating $1/\mathbb{P}(e = 1 \mid X = x)$.
**Procedure:**
1: **for** $\tau$ in a grid over $[0, 1]$ **do**
2:     Compute the fused miscoverage estimator:

$$\hat{\alpha}_{\text{focus}}(\tau) = \frac{\sum_{i \in \mathcal{I}_{\text{cal}}} \hat{w}(X_i) \cdot \mathbb{I}\{e_i = 1\} \cdot \mathbb{I}\{\tilde{T}_i < \hat{q}_\tau(X_i)\}}{\sum_{i \in \mathcal{I}_{\text{cal}}} \hat{w}(X_i) \cdot \mathbb{I}\{e_i = 1\}}$$

3: **end for**
4: Calibrate the hyperparameter $\tau$:

$$\hat{\tau}_{\text{focus}} = \sup\left\{\tau \in [0, 1] : \sup_{\tau' \leq \tau} \hat{\alpha}_{\text{focus}}(\tau') \leq \alpha\right\}$$

5: **Return:** The calibrated LPB: $\hat{L}(\cdot) := \hat{L}(\cdot; \hat{\tau}_{\text{focus}}) = \hat{q}_{\hat{\tau}_{\text{focus}}}(\cdot)$

---

**Algorithm 3** Conformalized survival analysis for general right-censored data: `fused` calibration

---

**Input:** desired miscoverage level $\alpha$; calibration data $\mathcal{D}_{\text{cal}} = \{(X_i, \tilde{T}_i, e_i)\}_{i \in \mathcal{I}_{\text{cal}}}$; estimated quantiles of $T \mid X$, $\{\hat{q}_\tau(\cdot)\}_{\tau \in [0,1]}$; weights $\hat{w}_\tau(x)$ approximating $1/\mathbb{P}(\zeta_\tau(X, e) = 1 \mid X = x)$, with the corresponding selection rule $\zeta_\tau : \mathcal{X} \times \{0, 1\} \to \{0, 1\}$ defined in Equation 9.
**Procedure:**
1: **for** $\tau$ in a grid over $[0, 1]$ **do**[2]
2:     Compute the fused miscoverage estimator:

$$\hat{\alpha}_{\text{fused}}(\tau) = \frac{\sum_{i \in \mathcal{I}_{\text{cal}}} \hat{w}_\tau(X_i) \cdot \mathbb{I}\{\zeta_\tau(X_i, e_i) = 1\} \cdot \mathbb{I}\{\tilde{T}_i < \hat{q}_\tau(X_i)\}}{\sum_{i \in \mathcal{I}_{\text{cal}}} \hat{w}_\tau(X_i) \cdot \mathbb{I}\{\zeta_\tau(X_i, e_i) = 1\}}$$

3: **end for**
4: Calibrate the hyperparameter $\tau$:

$$\hat{\tau}_{\text{fused}} = \sup\left\{\tau \in [0, 1] : \sup_{\tau' \leq \tau} \hat{\alpha}_{\text{fused}}(\tau') \leq \alpha\right\}$$

5: **Return:** The calibrated LPB: $\hat{L}(\cdot) := \hat{L}(\cdot; \hat{\tau}_{\text{fused}}) = \hat{q}_{\hat{\tau}_{\text{fused}}}(\cdot)$

---

# D    PROOFS FOR FUSED CALIBRATION DOUBLE ROBUSTNESS

## D.1    FUSED CALIBRATION THEORETICAL DERIVATION AND LEMMA

The derivation for the `fused` calibration method is very similar to that of `focused` calibration, with $\zeta_\tau(X, e)$ replacing $e$:

$$\alpha(\tau) = \mathbb{P}\big(T < \hat{q}_\tau(X)\big) = \mathbb{E}\left[\mathbb{P}(T < \hat{q}_\tau(X) \mid X)\right] \tag{11}$$
$$= \mathbb{E}\left[\mathbb{P}(T < \hat{q}_\tau(X) \mid X) \cdot \mathbb{P}(\zeta_\tau(X, e) = 1 \mid X) \cdot w(X)\right]$$
$$\leq \mathbb{E}\left[\mathbb{P}(T < \hat{q}_\tau(X), \zeta_\tau(X, e) = 1 \mid X) \cdot w(X)\right] \tag{12}$$
$$= \mathbb{E}\left[\mathbb{I}\{T < \hat{q}_\tau(X), \zeta_\tau(X, e) = 1\} \cdot w(X)\right] \tag{13}$$
$$= \mathbb{E}\left[\mathbb{I}\{\tilde{T} < \hat{q}_\tau(X)\} \cdot \mathbb{I}\{\zeta_\tau(X, e) = 1\} \cdot w(X)\right] = \alpha_{\text{focus}}(\tau).$$

---

[2] Observe that the values of $\tau$ that lead to a change in $\hat{\alpha}(\tau)$ correspond to those for which $\tilde{T}_i = \hat{q}_\tau(X_i)$, for $i \in \mathcal{I}_{\text{cal}}$. Consequently, the grid is defined over the $\tau$ values that satisfy this equality, with $\tau = 0$ included.

Where this time $w(x) = 1/\mathbb{P}(\zeta_\tau(X, e) = 1 \mid X = x) = \min(1/\mathbb{P}(e = 1 \mid X = x), s_\tau(X))$, steps Equation 11 and Equation 13 hold by the tower property, and step Equation 12 holds by the following lemma -

**Lemma D.1.** *Let* $s : \mathcal{X} \to \{0, 1\}$, *and* $\zeta_\tau : \mathcal{X} \times \{0, 1\} \to \{0, 1\}$ *as defined in Equation 9. Then Under Assumtion 1.1,* $\forall x \in \mathcal{X}, q : \mathcal{X} \to \mathbb{R}^+$,

$$\mathbb{P}[\zeta_\tau(X, e) = 1, \tilde{T} < q(X) \mid X = x] - \mathbb{P}[\zeta_\tau(X, e) = 1 \mid X = x]\mathbb{P}[T < q(X) \mid X = x] \geq 0.$$

*Proof.* We'll prove this lemma by separating the cases into the two possible values of $\hat{s}_\tau(X)$, which is constant given $X = x$. Suppose that $x \in \mathcal{X}$ satisfies $\hat{s}_\tau(x) = 0$. We have that

$$\mathbb{P}[\zeta_\tau(X, e) = 1, \tilde{T} < q(X) \mid X = x]$$
$$= \mathbb{P}[e = 1, \tilde{T} < q(X) \mid X = x]$$
$$= \mathbb{P}[e = 1, T < q(X) \mid X = x],$$

and

$$\mathbb{P}[\zeta_\tau(X, e) = 1 \mid X = x]\mathbb{P}[T < q(X) \mid X = x]$$
$$= \mathbb{P}[e = 1 \mid X = x]\mathbb{P}[T < q(X) \mid X = x].$$

And so, by Lemma B.1, we have that

$$\mathbb{P}[\zeta_\tau(X, e) = 1, \tilde{T} < q(X) \mid X = x] - \mathbb{P}[\zeta_\tau(X, e) = 1 \mid X = x]\mathbb{P}[T < q(X) \mid X = x]$$
$$= \mathbb{P}[e = 1, T < q(X) \mid X = x] - \mathbb{P}[e = 1 \mid X = x]\mathbb{P}[T < q(X) \mid X = x] \geq 0.$$

For $x \in \mathcal{X}$ with $\hat{s}_\tau(X) = 1$, we have that

$$\mathbb{P}[\zeta_\tau(X, e) = 1, \tilde{T} < q(X) \mid X = x]$$
$$= \mathbb{P}[\tilde{T} < q(X) \mid X = x],$$

And

$$\mathbb{P}[\zeta_\tau(X, e) = 1 \mid X = x]\mathbb{P}[T < q(X) \mid X = x]$$
$$= \mathbb{P}[T < q(X) \mid X = x].$$

And so, since by definition $\tilde{T} \leq T$, we have that $\forall x \in \mathcal{X}$:

$$\mathbb{P}[\zeta_\tau(X, e) = 1, \tilde{T} < q(X) \mid X = x] - \mathbb{P}[\zeta_\tau(X, e) = 1 \mid X = x]\mathbb{P}[T < q(X) \mid X = x]$$
$$= \mathbb{P}[\tilde{T} < q(X) \mid X = x] - \mathbb{P}[T < q(X) \mid X = x] \geq 0.$$

$\square$

### D.2  FUSED CALIBRATION WITH APPROXIMATELY ACCURATE WEIGHTS

In this section we prove Theorem 3.1, relying on the proof of (Gui et al., 2024, Theorem 3).

*Proof.* Here, we provide a single proof for the validity of both the `focused` and `fused`, using the appropriate notations for each method. For the `focused` method, we consider $\hat{\alpha}(\cdot) := \hat{\alpha}_{\text{focused}}(\cdot)$, $\hat{\tau} := \hat{\tau}_{\text{focused}}$, and $\zeta_\lambda(X_i, e_i) = e_i$. For the `fused` method, we denote $\hat{\alpha}(\cdot) := \hat{\alpha}_{\text{fused}}(\cdot)$, $\hat{\tau} := \hat{\tau}_{\text{fused}}$, and $\zeta_\lambda(X_i, e_i) = \mathbb{I}\{s_\lambda(X_i) = 1 \text{ or } e_i = 1\}$. Importantly, in both cases, the weights are defined as $w_\tau(x) := 1/\mathbb{P}(\zeta_\lambda(X, e) = 1 \mid X = x)$. We remark that all our claims hold for each of the two choices of these terms.

We define the error term by:

$$\Delta := \sup_{\lambda \in [0,1]} \left\{ \mathbb{E}\left[\left|\frac{\hat{w}_\lambda(X)}{w_\lambda(X)\pi_\lambda} - 1\right| \Big| \mathcal{D}_{\text{tr}}\right] + \sqrt{\frac{1 + \frac{\hat{\gamma}_\lambda^2}{\pi_\lambda^2} + \max\left(1, \frac{\hat{\gamma}_\lambda}{\pi_\lambda} - 1\right)^2}{0.4|\mathcal{I}_{\text{cal}}|} \cdot \log\left(\frac{1}{\delta}\right)} \right\}$$

Recall that the oracle quantity is formulated as:

$$\tau(\alpha + \Delta) = \sup\left\{\lambda \in [0,1] : \mathbb{P}\left(T < \hat{q}_\lambda(X) \mid \mathcal{D}_{\mathrm{tr}}\right) \le \alpha + \Delta\right\}.$$

The outline of this proof builds on the proof of (Gui et al., 2024, Theorem 3). First, if $1 - \delta \le \mathbb{P}(\hat{\tau} \le \tau(\alpha + \Delta) \mid \mathcal{D}_{\mathrm{tr}})$, then the event $\{\hat{\tau} \le \tau(\alpha + \Delta)\}$ holds with probability at least $1 - \delta$. Therefore:

$$\begin{aligned}
&\mathbb{P}(T \ge \hat{q}_{\hat{\tau}} \mid \mathcal{D}) \\
\ge&\mathbb{P}(T \ge \hat{q}_{\tau(\alpha+\Delta)}(X) \mid \mathcal{D}) \\
\ge&1 - \alpha - \Delta.
\end{aligned}$$

Above, the first inequality is due to the monotonicity of $\hat{q}_\tau$ and the second one follows from the left-continuity of $\mathbb{P}(T \ge \hat{q}_\tau(X) \mid \mathcal{D})$ in $\tau$. In what follows, we focus on showing $1 - \delta \le \mathbb{P}(\hat{\tau} \le \tau(\alpha + \Delta) \mid \mathcal{D}_{\mathrm{tr}})$. Suppose that $\varepsilon > 0$. For simplicity, we denote $\lambda := \tau(\alpha + \Delta) + \varepsilon$. Following the definition of $\hat{\alpha}(\tau)$, we get:

$$\begin{aligned}
&\mathbb{P}(\hat{\alpha}(\tau(\alpha + \Delta) + \varepsilon) \le \alpha \mid \mathcal{D}_{\mathrm{tr}}) \\
=&\mathbb{P}(\hat{\alpha}(\lambda) \le \alpha \mid \mathcal{D}_{\mathrm{tr}}) \\
=&\mathbb{P}\left(\frac{\sum_{i \in \mathcal{I}_{\mathrm{cal}}} \hat{w}_\lambda(X_i)\, \zeta_\lambda(X_i, e_i) \mathbb{I}\{\tilde{T}_i < \hat{q}_\lambda(X_i)\}}{\sum_{i \in \mathcal{I}_{\mathrm{cal}}} \hat{w}_\lambda(X_i)\, \zeta_\lambda(X_i, e_i)} \le \alpha \middle| \mathcal{D}_{\mathrm{tr}}\right) \\
=&\mathbb{P}\left(\sum_{i \in \mathcal{I}_{\mathrm{cal}}} \hat{w}_\lambda(X_i)\, \zeta_\lambda(X_i, e_i) \mathbb{I}\{\tilde{T}_i < \hat{q}_\lambda(X_i)\} \le \alpha \sum_{i \in \mathcal{I}_{\mathrm{cal}}} \hat{w}_\lambda(X_i)\, \zeta_\lambda(X_i, e_i) \middle| \mathcal{D}_{\mathrm{tr}}\right) \quad (14) \\
=&\mathbb{P}\left(\sum_{i \in \mathcal{I}_{\mathrm{cal}}} \hat{w}_\lambda(X_i)\, \zeta_\lambda(X_i, e_i) \left[\mathbb{I}\{\tilde{T}_i < \hat{q}_\lambda(X_i)\} - \alpha\right] \le 0 \middle| \mathcal{D}_{\mathrm{tr}}\right) \\
=&\mathbb{P}\left(\sum_{i \in \mathcal{I}_{\mathrm{cal}}} \hat{w}_\lambda(X_i)\, \zeta_\lambda(X_i, e_i) \left[\alpha - \mathbb{I}\{\tilde{T}_i < \hat{q}_\lambda(X_i)\}\right] \ge 0 \middle| \mathcal{D}_{\mathrm{tr}}\right)
\end{aligned}$$

Next, we apply Markov's inequality for any $t > 0$, and have:

$$\begin{aligned}
14 \le &\mathbb{E}\left(\exp\left(t \sum_{i \in \mathcal{I}_{\mathrm{cal}}} \hat{w}_\lambda(X_i)\, \zeta_\lambda(X_i, e_i)\left[\alpha - \mathbb{I}\{\tilde{T}_i < \hat{q}_\lambda(X_i)\}\right]\right)\middle|\mathcal{D}_{\mathrm{tr}}\right) \\
= &\mathbb{E}\left(\exp\left(t \sum_{i \in \mathcal{I}_{\mathrm{cal}}} \hat{w}_\lambda(X_i)\, \zeta_\lambda(X_i, e_i)\left[\alpha + p_\lambda(X_i) - p_\lambda(X_i) - \mathbb{I}\{\tilde{T}_i < \hat{q}_\lambda(X_i)\}\right]\right)\middle|\mathcal{D}_{\mathrm{tr}}\right).
\end{aligned}$$
$$(15)$$

Above $p_\lambda(x) := \mathbb{P}(T < \hat{q}_\lambda(X) \mid X = x, \mathcal{D}_{\mathrm{tr}})$. We now conditioning on $(X_i, e_i)_{i \in \mathcal{I}_{\mathrm{cal}}}$

$$\begin{aligned}
&\mathbb{E}\left(\exp\left(t \sum_{i \in \mathcal{I}_{\mathrm{cal}}} \hat{w}_\lambda(X_i)\, \zeta_\lambda(X_i, e_i)\left[p_\lambda(X_i) - \mathbb{I}\{\tilde{T}_i < \hat{q}_\lambda(X_i)\}\right]\right)\middle|\mathcal{D}_{\mathrm{tr}}, (X_i, e_i)_{i \in \mathcal{I}_{\mathrm{cal}}}\right) \\
\overset{(a)}{\le}&\exp\left(t\mathbb{E}\left(\sum_{i \in \mathcal{I}_{\mathrm{cal}}} \hat{w}_\lambda(X_i)\, \zeta_\lambda(X_i, e_i)\left[p_\lambda(X_i) - \mathbb{I}\{\tilde{T}_i < \hat{q}_\lambda(X_i)\}\right]\middle|\mathcal{D}_{\mathrm{tr}}, (X_i, e_i)_{i \in \mathcal{I}_{\mathrm{cal}}}\right) + \frac{t^2 \sum_{i \in \mathcal{I}_{\mathrm{cal}}} (2\hat{w}_\lambda(X_i))^2}{8}\right) \\
\overset{(b)}{\le}&\exp\left(t\mathbb{E}\left(\sum_{i \in \mathcal{I}_{\mathrm{cal}}} \hat{w}_\lambda(X_i)\, \zeta_\lambda(X_i, e_i)\left[p_\lambda(X_i) - \mathbb{I}\{\tilde{T}_i < \hat{q}_\lambda(X_i)\}\right]\middle|\mathcal{D}_{\mathrm{tr}}, (X_i, e_i)_{i \in \mathcal{I}_{\mathrm{cal}}}\right) + \frac{|\mathcal{I}_{\mathrm{cal}}| t^2 \hat{\gamma}_\lambda^2}{2}\right) \\
\le&\exp\left(t\mathbb{E}\left(\sum_{i \in \mathcal{I}_{\mathrm{cal}}} \hat{w}_\lambda(X_i)\left[\zeta_\lambda(X_i, e_i) p_\lambda(X_i) - \zeta_\lambda(X_i, e_i)\mathbb{I}\{\tilde{T}_i < \hat{q}_\lambda(X_i)\}\right]\middle|\mathcal{D}_{\mathrm{tr}}, (X_i, e_i)_{i \in \mathcal{I}_{\mathrm{cal}}}\right) + \frac{|\mathcal{I}_{\mathrm{cal}}| t^2 \hat{\gamma}_\lambda^2}{2}\right) \\
\overset{(c)}{\le}&\exp\left(t\left(\sum_{i \in \mathcal{I}_{\mathrm{cal}}} \hat{w}_\lambda(X_i)\,\mathbb{E}\left[\zeta_\lambda(X_i, e_i) p_\lambda(X_i) - \zeta_\lambda(X_i, e_i)\mathbb{I}\{\tilde{T}_i < \hat{q}_\lambda(X_i)\}\middle|\mathcal{D}_{\mathrm{tr}}, X_i, e_i\right]\right) + \frac{|\mathcal{I}_{\mathrm{cal}}| t^2 \hat{\gamma}_\lambda^2}{2}\right),
\end{aligned}$$
$$(16)$$

where step (a) uses Hoeffding's inequality; step (b) follows from the boundness of $\hat{w}$; step (c) is due to the independence assumption between the samples. We now turn to develop the expectation inside the sum.

$$\mathbb{E}\left[\zeta_\lambda(X_i, e_i)p_\lambda(X_i) - \zeta_\lambda(X_i, e_i)\mathbb{I}\{\tilde{T}_i < \hat{q}_\lambda(X_i)\} \mid \mathcal{D}_{\text{tr}}, X_i, e_i\right]$$

$$=\mathbb{E}\left[\zeta_\lambda(X_i, e_i) \mid \mathcal{D}_{\text{tr}}, X_i, e_i\right]\mathbb{E}\left[p_\lambda(X_i) \mid \mathcal{D}_{\text{tr}}, X_i, e_i\right] - \mathbb{E}\left[\zeta_\lambda(X_i, e_i)\mathbb{I}\{\tilde{T}_i < \hat{q}_\lambda(X_i)\} \mid \mathcal{D}_{\text{tr}}, X_i, e_i\right]$$

$$=\mathbb{P}\left[\zeta_\lambda(X_i, e_i) = 1 \mid \mathcal{D}_{\text{tr}}, X_i, e_i\right]\mathbb{E}\left[p_\lambda(X_i) \mid \mathcal{D}_{\text{tr}}, X_i, e_i\right] - \mathbb{P}\left[\zeta_\lambda(X_i, e_i) = 1, \tilde{T}_i < \hat{q}_\lambda(X_i) \mid \mathcal{D}_{\text{tr}}, X_i, e_i\right]$$

$$=\mathbb{P}\left[\zeta_\lambda(X_i, e_i) = 1 \mid \mathcal{D}_{\text{tr}}, X_i, e_i\right]\mathbb{E}\left[p_\lambda(X_i) \mid \mathcal{D}_{\text{tr}}, X_i, e_i\right] - \mathbb{P}\left[\zeta_\lambda(X_i, e_i) = 1, \tilde{T}_i < \hat{q}_\lambda(X_i) \mid \mathcal{D}_{\text{tr}}, X_i, e_i\right]$$

$$=\mathbb{P}\left[\zeta_\lambda(X_i, e_i) = 1 \mid \mathcal{D}_{\text{tr}}, X_i, e_i\right]\mathbb{P}(T_i < \hat{q}_\lambda(X_i) \mid \mathcal{D}_{\text{tr}}, X_i, e_i)$$
$$- \mathbb{P}\left(\zeta_\lambda(X_i, e_i) = 1 \mid \mathcal{D}_{\text{tr}}, X_i, e_i\right)\mathbb{P}\left(\tilde{T}_i < \hat{q}_\lambda(X_i) \mid \mathcal{D}_{\text{tr}}, X_i, e_i\right)$$

$$=\mathbb{P}\left[\zeta_\lambda(X_i, e_i) = 1 \mid \mathcal{D}_{\text{tr}}, X_i, e_i\right]\left[\mathbb{P}(T_i < \hat{q}_\lambda(X_i) \mid \mathcal{D}_{\text{tr}}, X_i, e_i) - \mathbb{P}\left(\tilde{T}_i < \hat{q}_\lambda(X_i) \mid \mathcal{D}_{\text{tr}}, X_i, e_i\right)\right]$$

$$\leq 0$$

The last inequality holds since $\tilde{T}_i \leq T_i$. By plugging this in we get: $16 \leq \exp\left(\frac{|\mathcal{I}_{\text{cal}}|t^2\hat{\gamma}_\lambda^2}{2}\right)$. We now condition 15 on $\{X_i, e_i\}_{i \in \mathcal{I}_{\text{cal}}}$:

$$\mathbb{E}\left(\exp\left(t\sum_{i \in \mathcal{I}_{\text{cal}}}\hat{w}_\lambda(X_i)\zeta_\lambda(X_i, e_i)\left[\alpha + p_\lambda(X_i) - p_\lambda(X_i) - \mathbb{I}\{\tilde{T}_i < \hat{q}_\lambda(X_i)\}\right]\right)\bigg|\mathcal{D}_{\text{tr}}, (X_i, e_i)_{i \in \mathcal{I}_{\text{cal}}}\right)$$

$$=\mathbb{E}\left(\exp\left(t\sum_{i \in \mathcal{I}_{\text{cal}}}\hat{w}_\lambda(X_i)\zeta_\lambda(X_i, e_i)\left[p_\lambda(X_i) - \mathbb{I}\{\tilde{T}_i < \hat{q}_\lambda(X_i)\}\right]\right.\right.$$
$$\left.\left. + t\sum_{i \in \mathcal{I}_{\text{cal}}}\hat{w}_\lambda(X_i)\zeta_\lambda(X_i, e_i)\left[\alpha - p_\lambda(X_i)\right]\right)\bigg|\mathcal{D}_{\text{tr}}, (X_i, e_i)_{i \in \mathcal{I}_{\text{cal}}}\right)$$

$$=\mathbb{E}\left(\exp\left(t\sum_{i \in \mathcal{I}_{\text{cal}}}\hat{w}_\lambda(X_i)\zeta_\lambda(X_i, e_i)\left[p_\lambda(X_i) - \mathbb{I}\{\tilde{T}_i < \hat{q}_\lambda(X_i)\}\right]\right)\right.$$
$$\left.\cdot \exp\left(t\sum_{i \in \mathcal{I}_{\text{cal}}}\hat{w}_\lambda(X_i)\zeta_\lambda(X_i, e_i)\left[\alpha - p_\lambda(X_i)\right]\right)\bigg|\mathcal{D}_{\text{tr}}, (X_i, e_i)_{i \in \mathcal{I}_{\text{cal}}}\right)$$

$$=\mathbb{E}\left(\exp\left(t\sum_{i \in \mathcal{I}_{\text{cal}}}\hat{w}_\lambda(X_i)\zeta_\lambda(X_i, e_i)\left[p_\lambda(X_i) - \mathbb{I}\{\tilde{T}_i < \hat{q}_\lambda(X_i)\}\right]\right)\bigg|\mathcal{D}_{\text{tr}}, (X_i, e_i)_{i \in \mathcal{I}_{\text{cal}}}\right)$$
$$\cdot \mathbb{E}\left(\exp\left(t\sum_{i \in \mathcal{I}_{\text{cal}}}\hat{w}_\lambda(X_i)\zeta_\lambda(X_i, e_i)\left[\alpha - p_\lambda(X_i)\right]\right)\bigg|\mathcal{D}_{\text{tr}}, (X_i, e_i)_{i \in \mathcal{I}_{\text{cal}}}\right)$$

$$\leq \exp\left(\frac{|\mathcal{I}_{\text{cal}}|t^2\hat{\gamma}_\lambda^2}{2}\right)\cdot\mathbb{E}\left(\exp\left(t\sum_{i \in \mathcal{I}_{\text{cal}}}\hat{w}_\lambda(X_i)\zeta_\lambda(X_i, e_i)\left[\alpha - p_\lambda(X_i)\right]\right)\bigg|\mathcal{D}_{\text{tr}}, (X_i, e_i)_{i \in \mathcal{I}_{\text{cal}}}\right)$$

By the law of total expectation, we get:

$$15 \leq \exp\left(\frac{|\mathcal{I}_{\text{cal}}|t^2\hat{\gamma}_\lambda^2}{2}\right)\cdot\mathbb{E}\left(\exp\left(t\sum_{i \in \mathcal{I}_{\text{cal}}}\hat{w}_\lambda(X_i)\zeta_\lambda(X_i, e_i)\left[\alpha - p_\lambda(X_i)\right]\right)\bigg|\mathcal{D}_{\text{tr}}\right) \tag{17}$$

We now condition on $\{X_i\}_{i\in\mathcal{I}_{\text{cal}}}$ and by the sub-gaussianity of $\zeta_\lambda(X_i, e_i) - w_\lambda(X_i)^{-1}$ we get:

$$\mathbb{E}\left(\exp\left(t\sum_{i\in\mathcal{I}_{\text{cal}}} \hat{w}_\lambda(X_i)\left(\zeta_\lambda(X_i,e_i) - w_\lambda(X_i)^{-1}\right)[\alpha - p_\lambda(X_i)]\right)\Bigg|\mathcal{D}_{\text{tr}}, \{X_i\}_{i\in\mathcal{I}_{\text{cal}}}\right)$$

$$\leq \exp\left(\frac{t^2}{8}\sum_{i\in\mathcal{I}_{\text{cal}}} \hat{w}_\lambda(X_i)^2 [\alpha - p_\lambda(X_i)]^2\right)$$

$$\leq \exp\left(\frac{t^2\hat{\gamma}_\lambda^2|\mathcal{I}_{\text{cal}}|}{8}\right)$$

We plug the above in 17 and bound it as follows:

$$17 \leq \exp\left(\frac{5|\mathcal{I}_{\text{cal}}|t^2\hat{\gamma}_\lambda^2}{8}\right) \cdot \mathbb{E}\left(\exp\left(t\sum_{i\in\mathcal{I}_{\text{cal}}} \frac{\hat{w}_\lambda(X_i)}{w_\lambda(X_i)}[\alpha - p_\lambda(X_i)]\right)\Bigg|\mathcal{D}_{\text{tr}}\right) \tag{18}$$

By combining all of the above, and by following the derivations in the proof of (Gui et al., 2024, Theorem 3) we get

$$18 \leq \exp\left(\frac{5|\mathcal{I}_{\text{cal}}|t^2}{8}\left(\pi_\lambda^2 + \hat{\gamma}_\lambda^2 + \tilde{\gamma}(\lambda)^2\right) + |\mathcal{I}_{\text{cal}}|t\left(\mathbb{E}\left[\left|\frac{\hat{w}_\lambda(X)}{w_\lambda(X)} - \pi_\lambda\right|\Big|\mathcal{D}_{\text{tr}}\right] - \pi_\lambda\Delta\right)\right) \tag{19}$$

By taking

$$t := \frac{\frac{4}{5}\left(\Delta\pi_\lambda - \mathbb{E}\left[\left|\frac{\hat{w}_\lambda(X)}{w_\lambda(X)} - \pi_\lambda\right|\Big|\mathcal{D}_{\text{tr}}\right]\right)}{\pi_\lambda^2 + \hat{\gamma}_\lambda^2 + \tilde{\gamma}(\lambda)^2},$$

we have

$$19 \leq \exp\left(-\frac{0.4|\mathcal{I}_{\text{cal}}|\left(\pi_\lambda\Delta - \mathbb{E}\left[\left|\frac{\hat{w}_\lambda(X)}{w_\lambda(X)} - \pi_\lambda\right|\Big|\mathcal{D}_{\text{tr}}\right]\right)^2}{\pi_\lambda^2 + \hat{\gamma}_\lambda^2 + \tilde{\gamma}(\lambda)^2}\right) \leq \delta,$$

where the last inequality follows from the definition of $\Delta$. Therefore,

$$1 - \delta \leq \mathbb{P}(\hat{\alpha}(\tau(\alpha + \Delta) + \varepsilon) > \alpha \mid \mathcal{D}_{\text{tr}}) \leq \mathbb{P}(\hat{\tau} < \tau(\alpha + \Delta) + \varepsilon \mid \mathcal{D}_{\text{tr}})$$

By taking $\varepsilon \to 0$, and by the continuity of the probability measure, we obtain $1 - \delta \leq \mathbb{P}(\hat{\tau} < \tau(\alpha + \Delta) + \varepsilon \mid \mathcal{D}_{\text{tr}})$, which concludes the proof. $\square$

### D.3 FUSED CALIBRATION WITH APPROXIMATELY ACCURATE QUANTILES

Following (Gui et al., 2024), instead of proving Theorem 3.2 directly, we provide a more general theorem that implies Theorem 3.2. Our proof relies on the proof of (Gui et al., 2024, Theorem 5).

**Theorem D.2.** *Fix any $\delta, \alpha \in (0, 1)$. Under the same conditions of Theorem 3.1, assume further that there exists a constant $r > 0$ such that:*

1. *For $P_X$-almost all $x$: $\sup_{\xi\in[\tau(\alpha),\tau(\alpha+r)+\psi]} w_\xi(x) \leq \gamma$ and $\sup_{\xi\in[\tau(\alpha),\tau(\alpha+r)+\psi]} \hat{w}_\xi(x) \leq \hat{\gamma}$ for some constants $\gamma, \hat{\gamma}, \psi \geq 0$*

2. *for any $\eta \in [0, r]$, for any $\beta \in [\alpha, \alpha + r]$, and for $P_X$-almost all $x$:*
$$\mathbb{P}(T < q_\beta(X) + \eta \mid X = x) \leq \beta + B\eta,$$
$$\mathbb{P}(T < q_\beta(X) - \eta \mid X = x) \geq \beta - B\eta.$$
   *for some family of oracle functions $\{q_\tau(\cdot)\}_{\tau\in[0,1]}$, and some constant $B > 0$.*

3. $\sup_{\beta\in[\alpha,\alpha+r]} \sup_{x\in\mathcal{X}} \left\{\max(B,1)|\hat{q}_{\tau(\beta)}(x) - q_\beta(x)| + \hat{\gamma}\gamma\sqrt{\frac{\log(1/\delta)}{0.4|\mathcal{I}_{\text{cal}}|}}\right\} \leq r.$

*Then with probability at least $1 - \delta$ over the draw of $\mathcal{D}$, the LPB produced by either Algorithm 3 or by Algorithm 2 satisfies that for $P_X$-almost all $x$:*

$$\mathbb{P}(T \geq \hat{L}(x) \mid \mathcal{D}, X = x) \geq 1 - \alpha - \sup_{\beta\in[\alpha,\alpha+r]}\sup_{x\in\mathcal{X}}\left\{2B|\hat{q}_{\tau(\beta)}(x) - q_\beta(x)|\right\} - \hat{\gamma}\gamma\sqrt{\frac{\log(1/\delta)}{0.4|\mathcal{I}_{\text{cal}}|}}.$$

*Proof.* Similarly to Appendix D.2, the following proof guarantees the validity of both the `focused` and `fused`. For each method, we embrace its appropriate notations as follows. For the `focused` method, we use $\hat{\alpha}(\cdot) := \hat{\alpha}_{\text{focused}}(\cdot)$, $\hat{\tau} := \hat{\tau}_{\text{focused}}$, and $\zeta_\lambda(X_i, e_i) = e_i$. For the `fused` method, we consider $\hat{\alpha}(\cdot) := \hat{\alpha}_{\text{fused}}(\cdot)$, $\hat{\tau} := \hat{\tau}_{\text{fused}}$, and $\zeta_\lambda(X_i, e_i) = \mathbb{I}\{s_\lambda(X_i) = 1 \text{ or } e_i = 1\}$. Notice that in both settings, the weights are formulated as $w_\tau(x) := 1/\mathbb{P}(\zeta_\lambda(X, e) = 1 \mid X = x)$. As in the proof in Appendix D.2, our claims hold for each choice of method and its corresponding terms.

We begin by defining the error terms:

$$\mathcal{E} = \sup_{\beta \in [\alpha, \alpha+r]} \sup_{x \in \mathcal{X}} \left\{ |\hat{q}_{\tau(\beta)}(x) - q_\beta(x)| \right\}, \quad \Delta = B\mathcal{E} + \hat{\gamma}\gamma \sqrt{\frac{\log(1/\delta)}{0.4|\mathcal{I}_{\text{cal}}|}}. \tag{20}$$

For simplicity, we also denote $\lambda := \tau(\alpha+\Delta)+\varepsilon$ for some $\varepsilon > 0$. We further adopt the definition of $p_a(x) := \mathbb{P}(T \leq \hat{q}_a(X) \mid X = x)$ from (Gui et al., 2024). The rest of the proof follows from (Gui et al., 2024, Theorem 5), except for substituting $\hat{f}$ by $\hat{q}$, $\mathbb{I}\{C_i \geq \hat{f}_\lambda(X_i)\}$ by $\zeta(X_i, e_i)$, $\mathbb{I}\{C_i \geq \hat{f}_\lambda(X_i) > T_i\}$ by $\mathbb{I}\{\zeta(X_i, e_i) = 1, \hat{f}_\lambda(X_i) > \tilde{T}_i\}$, and $\mathbb{I}\{T_i < \hat{f}_\lambda(X_i)\}$ by $\mathbb{I}\{\tilde{T}_i < \hat{f}_\lambda(X_i)\}$. All derivations from the proof of (Gui et al., 2024, Theorem 5) apply to our setting except for the upper bound of the following term, which we develop next.

$$\mathbb{E}\left[\exp\left\{t \sum_{i \in \mathcal{I}_{\text{cal}}} \frac{\hat{w}_\lambda(X_i)}{w_\lambda(X_i)} \left(p_\lambda(X_i) - \mathbb{I}\{\tilde{T}_i < \hat{q}_\lambda(X_i)\}\right) \middle| \mathcal{D}_{\text{tr}}, \{X_i\}_{i \in \mathcal{I}_{\text{cal}}}\right\}\right]$$

$$\leq \exp\left\{\frac{t^2}{2} \sum_{i \in \mathcal{I}_{\text{cal}}} \frac{\hat{w}_\lambda(X_i)^2}{w_\lambda(X_i)^2} + t \sum_{i \in \mathcal{I}_{\text{cal}}} \frac{\hat{w}_\lambda(X_i)}{w_\lambda(X_i)} \mathbb{E}\left[\left(p_\lambda(X_i) - \mathbb{I}\{\tilde{T}_i < \hat{q}_\lambda(X_i)\}\right) \middle| \mathcal{D}_{\text{tr}}, \{X_i\}_{i \in \mathcal{I}_{\text{cal}}}\right]\right\}$$

$$\leq \exp\left\{\frac{t^2}{2} \sum_{i \in \mathcal{I}_{\text{cal}}} \frac{\hat{w}_\lambda(X_i)^2}{w_\lambda(X_i)^2} + t \sum_{i \in \mathcal{I}_{\text{cal}}} \frac{\hat{w}_\lambda(X_i)}{w_\lambda(X_i)} \left[\mathbb{P}(T_i < \hat{q}_\lambda(X_i)|\mathcal{D}_{\text{tr}}, X_i) - \mathbb{P}(\tilde{T}_i < \hat{q}_\lambda(X_i)|\mathcal{D}_{\text{tr}}, X_i)\right]\right\}$$

$$\leq \exp\left\{\frac{t^2}{2} |\mathcal{I}_{\text{cal}}| \hat{\gamma}^2\right\}$$

Above, the first inequality follows from Hoeffding's inequality and the last one holds since $\tilde{T}_i \leq T_i$, and due to the upper bounds of the weights $\hat{w}_\lambda(\cdot)$. We plug in this bound in the derivations of (Gui et al., 2024) and get that

$$\mathbb{P}(\hat{\alpha}(\tau(\alpha+\Delta)+\varepsilon) > \alpha \mid \mathcal{D}_{\text{tr}}) \leq \exp\left\{-\frac{|\mathcal{I}_{\text{cal}}|t(\Delta - B\mathcal{E})}{\gamma} + \frac{5|\mathcal{I}_{\text{cal}}|\hat{\gamma}^2 t^2}{8}\right\}$$

Therefore, we take $t = \frac{4}{5\gamma\hat{\gamma}^2}(\Delta - B\mathcal{E})$ we get

$$\mathbb{P}(\hat{\alpha}(\tau(\alpha+\Delta)+\varepsilon) > \alpha \mid \mathcal{D}_{\text{tr}}) \leq \exp\left\{-0.4|\mathcal{I}_{\text{cal}}|\frac{1}{\gamma^2\hat{\gamma}^2}(\Delta - B\mathcal{E})^2\right\} \leq \delta$$

where the last inequality follows from the definition of $\Delta$ from Equation 20. Therefore, we get that with probability $1-\delta$ over the draw of $\mathcal{D}$: $\mathbb{P}(\hat{\alpha}(\tau(\alpha+\Delta)+\varepsilon) > \alpha)$. This implies $\tau(\alpha+\Delta)+\varepsilon > \hat{\tau}$, and thus:

$$\mathbb{P}(\tau(\alpha+\Delta)+\varepsilon > \hat{\tau}) \geq 1-\delta.$$

As in (Gui et al., 2024), we take $\varepsilon \to 0$, and by the continuity of the probability measure, we get that $\tau(\alpha+\Delta)+\varepsilon > \hat{\tau}$ holds with probability at least $1-\delta$, which concludes the proof.

$\square$

# E   DATASET DETAILS

## E.1   SYNTHETIC DATASETS

In all six simulated datasets from Section 4.1, the covariates $X$ are drawn from a uniform distribution, $X \sim U[0, 4]^p$. The base conditional survival time $T_{\text{base}} \mid X$ follows a lognormal distribution, as in Candès et al. (2023); Gui et al. (2024):

$$T_{\text{base}} \mid X \sim \exp(\mathcal{N}(\mu(X), \sigma^2(X))),$$

where $\mathcal{N}(\mu(X), \sigma^2(X))$ is the normal distribution with mean $\mu(X)$ and standard deviation $\sigma(X)$ defined in Table 1 for each setting, along with the dimension $p$ of $X$. To simulate patients with random early mortality, we choose 5% of subjects, and assign them the parameters $\mu = \ln 0.1$ and $\sigma = 0.2$ instead. For the remaining 95% event time is defined as $T = T_{\text{base}}$.

The censorship time $C \mid X$ is designed to mimic real-world censorship scenarios commonly encountered in clinical trials (Zhou et al., 2020; Wilson et al., 2021):

- **End-of-trial censorship**: All settings impose censorship at time 10.
- **Early censorship**: For 12% of subjects with $X[0] \in [0, 0.5]$, very early censorship is applied at $C = 0.1$, representing scenarios such as early loss to follow-up, treatment non-compliance, or withdrawal of consent.
- **Parametric follow-up censorship**: The remaining subjects are censored based on the parametric distribution $C_{\text{follow-up}}(x)$, described in Table 1, which simulates loss to follow-up over time.

Table 2 summarizes the overall censorship rates across different settings, ranging 30%-71.6%.

Table 1: Parameters specifying the distribution of survival time $T|X$ and censoring time $C|X$ in the six simulated data sets.

| Setting | $p$ | $\mu(x)$ | $\sigma(x)$ | $C_{\text{follow-up}}(x)$ |
|---|---|---|---|---|
| 1 | 4 | $\sqrt{x_1}$ | 1 | $\text{Exp}(5)$ |
| 2 | 4 | $(2 - \sqrt{x_4}) \cdot \mathbb{I}\{x_2 > 2\} + \sqrt{x_4} \cdot \mathbb{I}\{x_2 \le 2\}$ | 2 | $\text{Exp}(10)$ |
| 3 | 10 | $(2 - \sqrt{x_5}) \cdot \mathbb{I}\{x_1 > 2\} + x_4 \cdot \mathbb{I}\{x_1 \le 2\}$ | 1.5 | $\text{Exp}\left(\frac{1}{0.01 + \frac{4 - x_1}{50}}\right)$ |
| 4 | 10 | $3 \cdot \mathbb{I}\{x_1 > 2\} + 1.5 x_4 \cdot \mathbb{I}\{x_1 \le 2\}$ | 1.5 | $\text{lognormal}\left(2 + \frac{2 - x_1}{50}, 1.5\right)$ |
| 5 | 20 | $0.126(x_1 + \min(x_3 x_5, x_2 x_4))$ | 2 | $\text{Exp}\left(\frac{1}{\frac{x_4}{40} + \frac{x_1}{40}}\right)$ |
| 6 | 20 | $0.126(x_1 + \sqrt{x_3 x_5}) + 1$ | $\frac{x_2 + 2}{2}$ | $\text{Exp}\left(\frac{1}{\frac{x_{10}}{40} + \frac{1}{20}}\right)$ |

Table 2: Censorship rates for each simulation setting.

| Setting | 1 | 2 | 3 | 4 | 5 | 6 |
|---|---|---|---|---|---|---|
| **Censorship Rate (%)** | 64.6% | 40.5% | 30.0% | 71.6% | 45.3% | 52.6% |

### E.2 REAL-WORLD DATASETS

What follows are descriptions for the datasets used in the experiments of Section 4.2

The Northern Alberta Cancer Dataset (NACD), as presented by Haider et al. (2020), includes data on 2,402 patients diagnosed with a range of cancers, such as lung, colorectal, head and neck, esophageal, stomach, and other types. After excluding patients with negative or zero survival times, the dataset is reduced to 2,396 patients.

The Rotterdam & German Breast Cancer Study Group (GBSG) dataset combines information from two sources: the Rotterdam tumor bank Foekens et al. (2000) and the German Breast Cancer Study Group Schumacher et al. (1994). The Rotterdam dataset includes records for 1,546 patients with node-positive breast cancer, while the GBSG dataset provides complete data for 686 patients.

The Molecular Taxonomy of Breast Cancer International Consortium (METABRIC) dataset, detailed by Curtis et al. (2012), provides survival data for breast cancer patients. It incorporates a wide array of features, including clinical traits, gene expression profiles, copy number variation (CNV) profiles, and single nucleotide polymorphism (SNP) genotypes. These features are derived from breast tumor samples collected during the METABRIC study.

The Study to Understand Prognoses, Preferences, Outcomes, and Risks of Treatment (SUPPORT) dataset, as described by Knaus et al. (1995), includes data from 9,105 participants. It was designed to investigate survival outcomes and clinical decision-making processes for critically ill hospitalized patients.

The customer churn prediction dataset (Churn) is designed to predict customer attrition (Fotso et al., 2019–present). For analysis, individuals censored at time 0 are excluded.

The Cancer Genome Atlas Breast Invasive Carcinoma (TCGA-BRCA) multi-modal dataset collection (Tomczak et al., 2015) is also used, and includes various clinical features such as age, cancer grade, and tumor size, along with pathological biomarkers, genetic data, and visual biopsy samples. To effectively estimate the conditional survival times from such diverse features, we designed dedicated preprocessing and training procedures for this data. These include the use of the GigaPath foundation model (Xu et al., 2024) to extract features from the visual biopsy samples, extraction of cancer grade from patients' medical reports, and more.

A detailed description of the entire processing pipeline can be found in Appendix F.

Table 3 summarizes the overall censorship rates across the different datasets.

Table 3: Censorship rates for each real-world dataset.

| Setting | TCGA | SUPPORT | METABRIC | churn | NACD | GBSG |
|---|---|---|---|---|---|---|
| **Censorship Rate (%)** | 86.6% | 32.0% | 42.1% | 53.4% | 36.6% | 43.2% |

## F  EXPERIMENTAL SETUP

In all experiments, the dataset was split into four parts: 60% for training, 20% for calibration, 10% for validation (used for early stopping), and 10% for testing to evaluate performance. Synthetic data was generated through distribution simulations, as outlined in Table 1, while the process for acquiring the TCGA data is described in Section F.3. The synthetic data generation function and the processed TCGA-BRCA dataset are available in the github repository. The TCGA-BRCA dataset was further normalized and imputed using the dataset pre-processing code by Ketenci et al. (2023), utilizing the pre-processing of Nagpal et al. (2021a) for data imputation and normalization.

In all experiments, we approximated the distribution of $T \mid X$ using the *DeepSurv* method (Katzman et al., 2018), implemented in the *pycox* package (Kvamme et al., 2019), implemented using a PyTorch MLP regressor with ReLU activation, early stopping (triggered after 5 epochs without improvement), and a training cycle of 1000 epochs. The Adam optimizer optimized the model with parameters $lr = 1e - 3$, $\beta_1 = 0.9$ and $\beta_2 = 0.999$, a batch size of 256, dropout layers with a rate of $p = 0.1$, batch normalization layers, and varying configurations of hidden layers, detailed in Table 4. These configurations were selected to be similar to those found in the PyCox notebooks, with the real-world datasets getting a deeper model to account for their more complex and interconnected nature.

Additionally, we employed *scikit-learn* (Pedregosa et al., 2011) to train a Random Forrest Classifiers with a max depths of $4$ and $2$, to estimate the weights $\hat{w}_\tau$ and the indicator $\hat{s}_\tau$ respectively. The higher max depth for estimating $\hat{w}_\tau$ is meant to make conditional calibration more accurate, promoting validity. In comparison, the lower max depth of $\hat{s}_\tau$ is there to reduce potential overfitting, that might lead to overly conservative estimators.

### F.1  MACHINE SPECIFICATIONS

The hardware and OS used for the experiments are as follows.

- **CPU:** AMD EPYC 7443 24-Core Processor
- **GPU:** NVIDIA RTX A6000
- **OS:** Ubuntu 20.04

Table 4: Parameters specifying the number of hidden layers of the MLP regressor employed for the *DeepSurv* survival analysis model in the synthetic and TCGA settings.

| Setting | Hidden layers |
|---|---|
| Synthetic settings | [5] |
| TCGA-BRCA | [32, 32] |

## F.2 COMPUTATIONAL EFFICIENCY

Note that the values of $\tau$ that cause a shift in $\hat{\alpha}(\tau)$ are those for which $\tilde{T}_i = \hat{q}_\tau(X_i)$, for $i \in \mathcal{I}_{cal}$. As a result, to compute the miscoverage estimator one need only check these values and $\tau = 0$. As such, the computational complexity of the calibration is $O(|\mathcal{I}_{cal}|)$, and so the brunt of computing is allocated towards the model training, both for the quantile and weight estimation.

## F.3 TCGA ACQUISITION AND PREPROCESSING DETAILS

The TCGA-BRCA cohort is accessible through the GDC portal. To create a dataset suitable for *DeepSurv*, we exclude patients without clinically relevant Estrogen Receptor (ER) status or scanned biopsies. We then merge several tables and select the following clinically significant features for model training, imputing the Progesteron Receptor (PR) value with the ER value, and the rest of the features with the value value $-1$:

- ER status $\in \{0, 1\}$
- PR status $\in \{0, 1\}$
- Her2 status $\in \{0, 1\}$
- Presence of infiltrating ductal carcinoma $\in \{0, 1\}$
- Presence of infiltrating lobular carcinoma $\in \{0, 1\}$
- PGR gene expression $\in [0, 1]$
- ESR1 gene expression $\in [0, 1]$
- ERBB2 gene expression as defined by Desmedt et al. (2008) $\in [0, 1]$
- Gender $\in \{0, 1\}$
- Age $\in \{26, \cdots, 90\}$
- Tamoxifen drug sensitivity $\in [0, 1]$
- Lapatinib drug sensitivity $\in [0, 1]$

Additionally, we incorporate the cancer grade of each patient by combing through their written medical reports. Finally, we process a single whole-slide H&E biopsy image for each patient, which can vary significantly in size, typically around 300,000 pixels. To identify the tissue regions, we apply Otsu's segmentation method. After segmentation, we use the *GigaPath* foundation model from Xu et al. (2024) to generate compact embedding vectors that capture the essential features of each biopsy. These embeddings are then further reduced to 3 dimensions using PCA (Principal Component Analysis). All of the above features are concatenated to create a covariate vector $X$ of size 19. The survival time $T$ is defined as the time to mortality, with the censored time $\tilde{T}$, and event indicator $e$ given by the dataset.

## G STUDYING THE EFFECTS OF CENSORSHIP RATE CHANGE ON THE CALIBRATION METHODS' PERFORMANCE

In this section, we examine the performance of the proposed calibration methods under varying levels of censorship. We test all three of our methods on Setting 3 from Section 4.1 for a variety of end-of-trial times. Figure 3 presents the empirical coverage rate and LPB obtained by each

technique. This ablation study demonstrates the superiority of the fused calibration method as compared to the focused and naive methods. We can see that in this setting the fused method archives a relatively tight, yet valid, LPB regardless of the censorship rate, while the focused method's performance depends much more on the censorship rate, and is similar to that of the fused method only with minimal censorship. The naive method underperforms throughout.

This hierarchy in performance teaches us important lessons about each method's reactivity to changes in censorship:

- The naive method has stable performance across the varied end of trial times since it calibrated the estimator to the $0.1$-th quantile of $\tilde{T}$. Since we changed only the upper quantiles of $\tilde{T}$ by bounding $C$'s value, the lower quantiles remain unchanged, and so the calibration output remains unchanged.

- The focused method on the other hand, calibrated the estimator to the $0.1$-th quantile of $\tilde{T} \mid e = 1$, i.e. $T \mid e = 1$. As the end of trial time increases, more and more high values of $T$ with $e = 1$ get included in our calibration set, and so all quantile levels get higher, making the focused method less conservative.

- The fused method manages to avoid the dichotomy of $\tilde{T}$ and $\tilde{T} \mid e = 1$ by selectively choosing examples that make it less conservative. The degree to which it succeeds in this task is surprising, and motivates the authors to delve deeper into the reasons for this success in future work.

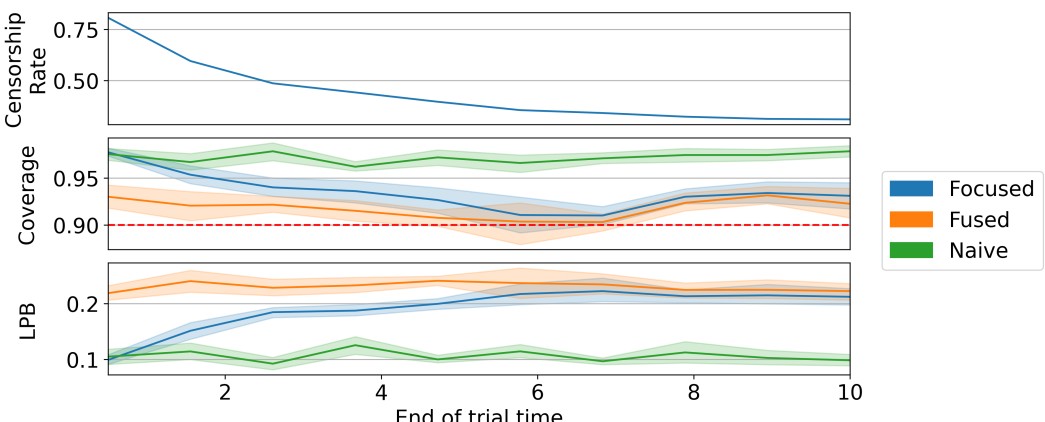

Figure 3: Performance of the naive, focused, and fused calibration methods as a function of end of trial time. **Top**: Censorship rate change. **Middle**: Empirical coverage rate, with a red dashed line indicating the nominal 90% level. **Bottom**: LPB. Higher is better. The reported performance metrics are evaluated on 10 independent trials, each consisting of newly sampled train, validation, calibration, and test sets of sizes $600, 200, 1000,$ and $200,$ respectively.

## H   RELATIVE PERFORMANCE OF THE METHODS AS A FUNCTION OF THE NUMBER OF SAMPLES

We demonstrate the effect of varying sample sizes in Setting 3 from Section 4.1 on the naive, focused, and fused calibration schemes. The coverage rate and LPB constructed by each method are presented in Figure 4. This figure indicates that all methods attain a valid coverage rate even when used with a small sample size.

We can see that when we use up to $800$ samples using a $50\%$ calibration set, both the focused and fused methods show relatively high uncertainty, although their performance is still superior to that of the naive method, and they maintain validity. Above the $800$ sample range, all methods perform about the same with a high sample number as they do with a lower sample number. We

can infer from these results that the validity of the calibration methods is unaffected by varying the number of samples, but the variance of the LPBs generated by the focused and fused methods, which don't always utilize all the calibration samples, gets predictably lower when calibrated using more samples.

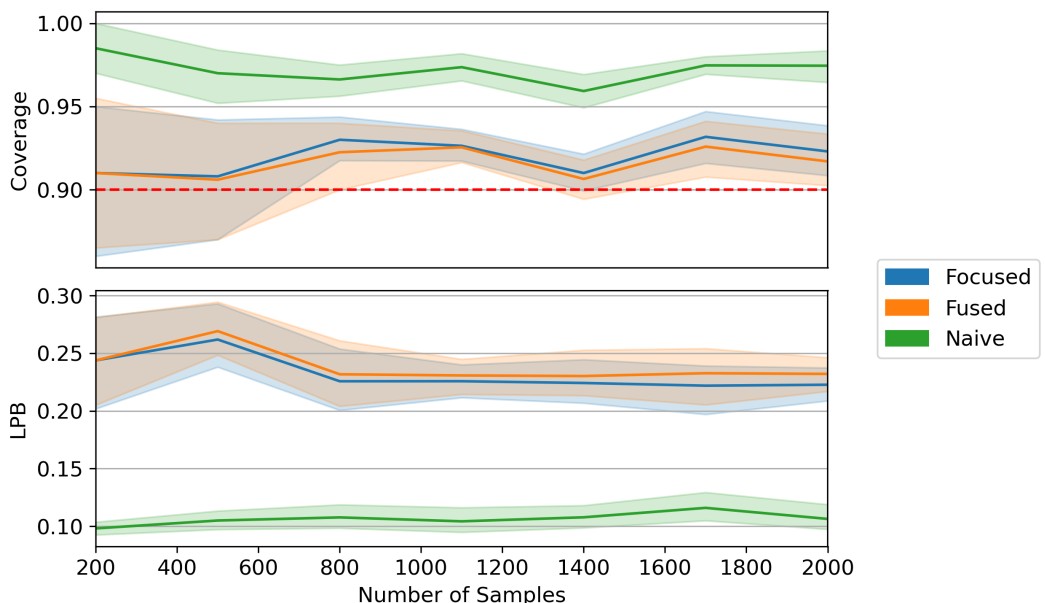

Figure 4: Performance of the naive, focused, and fused calibration methods as a function of sample size. **Top**: empirical coverage rate, with a red dashed line indicating the nominal 90% level. **Bottom**: LPB. A higher LPB is better. The reported performance metrics are evaluated on 10 independent trials, each consisting of newly sampled train, validation, calibration, and test sets which consist of $30\%, 10\%, 50\%,$ and $10\%$ proportions of the dataset, respectively.

# I  ABLATION STUDY: ROBUSTNESS TO WEIGHT AND QUANTILE ESTIMATION HYPERPARAMETERS

For weight estimation, we test the methods detailed in the paper using either a shallow random forest classifier (max depth 2) or a deeper one (max depth 6), and for conditional quantile estimation, we test the methods using either a shallow network (1 hidden layer) or a deeper network (3 hidden layers).

We analyze the performance of the proposed calibrations under this ablation, and display the coverage rate and LPB in Figure 5. This figure shows that our methods are robust to arbitrary selection of these hyperparameters in terms of both coverage validity and informativeness, and reinforce the superiority of the fused calibration method, which manages to perform on par or better than both the focused and the naive methods.

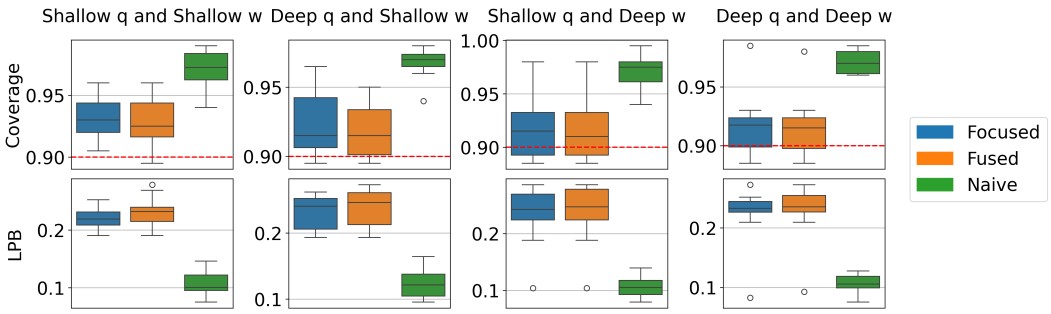

Figure 5: Performance of the naive, focused, and fused calibration methods for each combination of hyperparameters. **Top**: empirical coverage rate, with a red dashed line indicating the nominal 90% level. **Bottom**: LPB. A higher LPB is better. The reported performance metrics are evaluated on 10 independent trials, each consisting of newly sampled train, validation, calibration, and test sets of sizes $600, 200, 1000,$ and $200,$ respectively.

