# OpenReview forum: "Conformalized Survival Analysis for General Right-Censored Data"
_ICLR.cc/2025/Conference — ICLR 2025 Poster_

### Official Review · Reviewer_bciN · 2024-10-18

**Soundness:** 1
**Presentation:** 2
**Contribution:** 1
**Rating:** 3
**Confidence:** 4

**Summary:**

The paper proposes a reliable lower predictive bound (LPB) for the survival time under the general right-censored data based on the work in Gui et al. (2024). The proposed LPB is a PCA-type with distribution-free finite-sample guarantee.

**Strengths:**

1.	The LPB is doubly robust by theorem.
2.	The paper extends the result in Gui et al. (2024) to the general right-censoring setting.

**Weaknesses:**

1.	I don't see a strong motivation for using the general right-censoring setting, which appears to be quite similar to the Type-I right-censoring setting. Other than two citations (which only mention the definition), the authors do not provide a clear explanation of why the general right-censoring setting is timely or important in real-world data.
2.	The entire framework builds on Gui et al. (2024), except for the miscoverage estimators. The proposed focus calibration and fused calibration use only part of the information in the calibration set, which may lead to a waste of information. Additionally, I have concerns about the sufficiency of the proposed calibrations in cases of a large imbalance between censored and uncensored data in the calibration set (e.g., when the censoring rate is high). Therefore, there are doubts regarding the novelty of the proposed method.
3.	Many parts of the paper are derived from Gui et al. (2024), including the main idea, the theoretical results, the simulations, and the proofs. Several sections are not clearly explained in this paper, requiring a read of Gui et al. (2024) to understand the context. For example, why is a PAC-type LPB considered instead of a marginally calibrated LPB? How does the PAC-type LPB feature in your method? As a result, phrase like "Notably, ..." in line 162-163 is vague and confusing.
4.	Section 3.2 presents a list of results without a clear interpretation of the underlying ideas. The nested notation further complicates the readability of the section.
5.	In your experiments, there is no comparison between your method and other methods, except the naïve LPB. The experimental section is mainly a collection of results without sufficient interpretations. For instance, the focused and fused LPBs are not consistently better than the naïve LPB in settings 4 and 6. In the real data experiment, the focused and fused LPB are not significantly better than the naïve LPB.
6.	The data used in your experiments does not reflect a general right-censoring setting.
7.	Some notations in the proof are clunky and confusing. For example, in the proof of Lemma B.1, $q=q(X)$ in your $\mathbb{P}_x(\cdot)$

is deterministic? In Section B.2, the form
$\mathbb{E}[\mathbb{P}(\tilde{T}<\hat{q}_{\tau}(X)|X=x,e=1)]$

does not make sense,
as $\hat{q}_{\tau}(X)$ is not random given $X=x$.

8. Some references are not in correct format. For example, the arxiv references should be in arxiv format.

**Questions:**

1.	How do you handle the extreme $\hat{\omega}$?
2.	How does the estimator $\hat{s}_{\tau}(x)$ affect your method?
3.	The fused and focused LPB may not as tight as the LPB in Gui et al. (2024), then how loss are they? Can you quantify?
4.	In what settings, the fused and focused LPB are not better than the naïve LPB?
5.	Why the authors not include the average LPB in the experiments?
6.	What is the coverage result of the real data experiment? Can you run experiments on other real data set? For example, the data used in Candes et al. (2023) and Gui et al. (2024).


Emmanuel Cande`s, Lihua Lei, and Zhimei Ren. Conformalized survival analysis. Journal of the Royal Statistical Society Series B: Statistical Methodology, 85(1):24–45, 2023.

Yu Gui, Rohan Hore, Zhimei Ren, and Rina Foygel Barber. Conformalized survival analysis with adaptive cut-offs. Biometrika, 111(2):459–477, 2024.

---

> ### Author Response · Authors · 2024-11-20
> **Response to Reviewer bciN - part 1/3**
>
> We appreciate your engagement, helpful suggestions, and valuable feedback. In what follows, we respond to the reviewer’s concerns in detail.
>
> ## Weaknesses
>
> > The importance of the right-censoring setting
>
> We kindly refer the reviewer to GLOBAL RESPONSE: CONTRIBUTIONS (Importance of reliable survival analysis under general right-censored data).
>
> > The novelty of our work in view of Gui et al. (2024)
>
> We kindly refer the reviewer to GLOBAL RESPONSE: CONTRIBUTIONS (Key novelty) for a discussion of the significant theoretical and technical advancements in our work.
>
> > Using only a subset of the calibration set
>
> We understand that the idea of using a subset of the calibration points to enhance performance is not intuitive at first. However, this approach can greatly improve statistical efficiency compared to the naive method that uses all the calibration points.
> Our work focuses on carefully selecting a subset of calibration points to produce more informative LPBs while maintaining validity. In Proposition 3.1, we characterize the condition under which such a selection of a subset of points (those whose $e=1$) will provably improve performance. Loosely speaking, this is because only a subset of samples—the uncensored ones (for which $e_i=1$)—reveals the target variable $T_i$ that we want to estimate; the other censored points (for which $e_i=0$) only present a lower bound on the unobserved $T_i$.
> We note that the reviewer’s comment is aligned with the motivation behind our flagship method—fused calibration—that offers a rigorous strategy to seize the most informative censored points in addition to the uncensored ones to further enhance statistical efficiency. Indeed, our experiments show that our proposals—especially the fused method—attain better performance than the naive approach despite utilizing less data.
>
> > Extreme censoring rates
>
> The theoretical bounds provided in Section 3.3 are valid regardless of the censoring rate. Please refer to GLOBAL RESPONSE: EXPERIMENTS (Robustness analysis to extreme censorship rates) for an experiment that validates our argument.
>
> > Marginal vs. PAC-type guarantee
>
> The key difference between a marginal LPB and a PAC-type LPB lies in whether one is interested in obtaining valid coverage conditional on the calibration set (PAC) or not (marginal). In other words, the advantage of a PAC-type guarantee is that it characterizes how the randomness in the choice of the calibration set affects the validity: this is reflected by the “$\delta$ term” in Theorem 3.1 and Theorem 3.2, which shrinks as the size of the calibration set increases.
> In our work, we employ a PAC-type guarantee, which aligns with the typical scenario in survival analysis where the user is given a specific calibration set (often of limited size) and aims to obtain valid results conditioned on this set.
>
> > Valid lower bound on $\tilde{T}$ leads to a valid lower bound on $T$
>
> We apologize if the sentence in lines 162-163 is confusing. For convenience, we repeat it here: “Notably, this $\hat{L}(X; {\hat{\tau}_{\text{naive}}})$ is a PAC-type LPB on $\tilde{T}$, and thus also a PAC-type LPB on $T$.”
> This statement relies on the fact that  $\tilde{T} \leq T$ by definition, and so in an event where $\hat{L}$ is a valid lower bound on $\tilde{T}$, it will also be a lower bound on $T$. This is the core argument as to why the naive method is valid. We’ll include this clarification in the text. Thank you again for helping us improve our writing.
>
> > Readability of Section 3.2
>
> We are uncertain about what the reviewer specifically refers to as “results” in Section 3.2. However, we will take this opportunity to clarify the underlying idea of the fused algorithm presented in that section.
> The motivating question behind the fused method is this: how can we select informative censored points in addition to the uncensored ones to gain statistical efficiency? This is in contrast with the focused method that only uses uncensored points for calibration.
> The mechanism through which we choose censored points to participate in the calibration process is the censored selection estimator $\hat s_\tau(x)$, motivated by Proposition 3.1 (Eq. 8). In turn, the fused calibration set should ideally include the censored points that violate Eq. 8 and the uncensored points. To this end, we define the selection rule $\zeta_\tau(X_i, e_i)$ as an “or” operator between the event indicator $e_i=1$ and the censored selection estimator $\hat s_\tau(x)=1$.
> We acknowledge that the definition of the selection rule $\zeta_\tau(X_i, e_i) $ in Section 3.2 was not clearly motivated, which may have led to confusion. We will implement the above discussion to the text to enhance clarity. We greatly appreciate your feedback!

---

> ### Author Response · Authors · 2024-11-20
> **Response to Reviewer bciN - part 2/3**
>
> > Comparison to other methods
>
> Initially, we did not compare our methods to others as no existing techniques provide finite-sample coverage guarantees in the general right-censoring setup. We agree with the reviewer that such a comparison can shed light on the importance of our contribution, especially if the uncalibrated model or an asymptotic calibration method would not obtain the required coverage in practice.
> In response, we now include comparisons to the uncalibrated model and to the method developed by Qi et al. (2024), namely the CSD method. We kindly refer the reviewer to our GLOBAL RESPONSE: EXPERIMENTS, showing that for some synthetic datasets, the uncalibrated model and asymptotic CSD produce LPBs that undercover the test time-to-event variables. This is in contrast with our methods, which always attain valid coverage.
>
> > Analysis of synthetic experiments from the initial submission and real data experiment
>
> Thank you for your thoughtful analysis of our experiments. Following the reviewer's comment, we discuss below the results of Settings 4 and 6 from the initial synthetic datasets and the TCGA dataset.
> Setting 4: As pointed out by the reviewer, the coverage rate of the fused method is similar to that of the naive approach. However, notice that the LPBs generated by the fused method are higher than those of the focused and naive methods.
> Setting 6: Here, all methods produce valid LPBs with similar values. Although the overlapping whiskers make it difficult to draw definitive conclusions about performance, we attribute the slight underperformance of the fused method to inaccuracies in estimating the ideal selection rule $s(x)$.
> TCGA dataset: The fused and focused methods yield LPBs higher than the naive approach by a small but noticeable margin. While the improvement may appear modest, providing reliable and more informative predictions with LPBs that are a few months higher can have a meaningful impact on decision-making for breast cancer patients.
> We will include a discussion along these lines in the text to improve the interpretability of our experiments, thank you for raising this point.
>
> > Experiments reflecting a general right-censoring setting
>
> We respectfully disagree with the reviewer and apologize for the confusion.
>
> **To be sure, all our experiments follow the general right-censoring setting.**
>
> The right-censoring framework refers to scenarios where, for each sample, we observe either the survival time $T$ or the censoring time $C$, but not both. In all experiments, the observed data consists of the feature vector $X$, the variable $\tilde{T} = \min(T, C)$, and an indicator variable $e$, specifying whether $\tilde{T}=T$ or $\tilde{T}=C$. We never observe both $T$ and $C$.
>
> > Confusing notations
>
> Indeed, $q(X)$ is deterministic given $X=x$. We will clarify this point in the revised manuscript.
> Regarding the formula $E[P(\tilde{T}<q_{\tau}(X)|X=x,e=1)]$, you are correct, there is a typo. The correct form is $E[P(\tilde{T}<q_{\tau}(X)|X,e=1)]$, as the expectation is taken over draws of $X$. Thank you so much for catching this typo! We will correct it in the revised version.
>
> > References in arXiv format
>
> Thank you for this comment, when appropriate, we will revise and correct the format of the references.
>
> > Report average LPB
>
> Our figures include box plots showing the distribution of the LPBs and their median. In the synthetic data, we presented the relative LPB to improve readability (as otherwise, the y-axis would have a different scale for each setting). Following this comment, we will also include the average LPB values. Please refer to Tables 3 and 4 in “rebuttal_experients.pdf” of the Supplementary Material.
>
> > Coverage rate in the real data experiment
>
> It is impossible to compute the coverage rate for real survival data because $T$ is unknown for censored subjects—this is precisely what makes the right-censoring setting complex and widely studied.
> > Additional real datasets
>
> We kindly refer the reviewer to GLOBAL RESPONSE: EXPERIMENTS (real data).
>
> **References**
>
> Shi-Ang Qi, Yakun Yu, and Russell Greiner. Conformalized Survival Distributions: A Generic Post-Process to Increase Calibration. ICML, 2024.

---

> ### Author Response · Authors · 2024-11-20
> **Response to Reviewer bciN - part 3/3**
>
> ## Questions
>
> > Extreme weights
>
> That’s an excellent question! Handling extreme weights is of great interest, and fits into the broader context of weighted conformal prediction and other statistical methods that leverage weights to address covariate or label shift (Tibshirani et al. (2019); Candes et al. (2023); Gui et al. (2024)).
> We anticipate facing such an issue when we observe an uncensored sample with $e=1$ although its estimated $\hat{P}[e=1|X=x]$ is extremely small. In our experiments, we haven’t encountered such an issue, but one approach to handling extreme $\hat{w}$ could be bounding the estimated probability $\hat{P}[e=1|X=x]$, as suggested by Candes et al. (2023).
> From a theoretical perspective, when the true probability $P[e=1|X=x]$ is very low, there is a trade-off between limiting the estimated weights $\hat{w}(x)$ and maintaining accurate weight estimation. This balance directly impacts the coverage guarantee established in Theorem 3.1. According to the theorem, under-coverage is affected by two key factors: (1) a high value of $\hat{\gamma}$, which bounds $\hat{w}(x)$; and (2) the discrepancy between $w(x)$ and its estimator $\hat{w}(x)$. This analysis sheds light on the impact of bounding the weights. Imposing strict bounds on $\hat{w}(x)$ would reduce $\hat{\gamma}$ but would also negatively impact the accuracy of $\hat{w}(x)$. We will include this discussion in the revised manuscript.
>
> > The tightness of the LPBs
>
> Intuitively, when the miscoverage estimator can fully utilize the conditional independence assumption, $T \perp C | X$, we expect to obtain tight coverage. This happens for the method developed by Gui et al. (2024), as they showed how to cast the problem as a calibration under covariate shift. Yet, this formulation requires access to $C_i$ for all observations $i=1,...,n$.
> In the general right-censored setting, it is impossible to fully utilize the conditional independence assumption for coverage estimation: we do not know the value of $C$ of all samples! Therefore, it is impossible to cast the problem as calibration under covariate shift anymore. Yet, we have shown that we can translate the problem to a “calibration under conservative covariate shift”: kindly refer to the inequality in Eq. 5 that builds on Lemma B.1. Hence, unless the transition in Eq. 5 holds in equality, we expect to obtain conservative coverage. This is a natural consequence of having less information.
> Following the reviewer’s comment, we seek to analytically quantify the extent to which our calibration method is conservative. We believe this can be done by leveraging the set of steps in the proof of Lemma B.1. We hope to include such a result in the revised manuscript.
>
> > In what situations would the LPB of the focused and fused calibrations underperform compared to the naive approach?
>
> The focused method can underperform the naive method in settings where the early censorship rate is low, as rigorously stated in Proposition 3.1.
> The fused method is expected to match or exceed the performance of both the naive and focused methods, as long as the selection rule $\hat{s}(x)$ is well estimated. This is because the goal of $\hat{s}(x)$ is to detect the censored observations that violate the condition in Proposition 3.1. In turn, the fused approach flexibly includes or excludes censored calibration points in a manner that yields more informative LPBs regardless of whether early censoring is rare or common, for example. This flexibility is demonstrated in Figure 5 in “rebuttel_experiments.pdf” attached to the supplementary material.
> Lastly, we anticipate the fused and naive methods to perform the same when the condition in Proposition 3.1 is violated for the vast majority of samples.
>
> > The estimator $\hat s_{\tau}(x)$
>
> As stated above, the goal of $\hat s_{\tau}(x)$ in the fused method is to flexibly include or exclude censored calibration points in a manner that yields more informative LPBs.
> It’s important to note that even a poorly estimated $\hat s_{\tau}(x)$ ensures valid coverage. This is because the weights of the fused method are defined as $w(x)=1/P[e=1 | X=x]$ if $\hat{s}(x)=0$ and $1$ otherwise. Notice that this condition does not involve the oracle selection rule $s_\tau(x)$, but solely its estimate $\hat s_\tau(x)$. Thus, the consequence of relying on a poor estimation $\hat{s}(x)$ is reduced efficiency, not invalid coverage.
> We will clarify these points in the revised text.
>
> **References**
>
> Ryan J Tibshirani, Rina Foygel Barber, Emmanuel Candes, and Aaditya Ramdas. Conformal prediction under covariate shift. NeurIPS, 2019.
>
> Emmanuel Candes, Lihua Lei, and Zhimei Ren. Conformalized survival analysis. JRSS Series B: Statistical Methodology, 85(1):24–45, 2023.
>
> Yu Gui, Rohan Hore, Zhimei Ren, and Rina Foygel Barber. Conformalized survival analysis with adaptive cut-offs. Biometrika, 111(2):459–477, 2024.

---

> > ### Comment · Reviewer_bciN · 2024-11-25
> >
> > Thank you for your detailed replies. I would also like to thank `Reviewer rRgu` for the comments on the general right-censoring setting. Your responses addressed some of my concerns related to your experiments and the general right-censoring setting. I have no doubt about the validity of your theoretical results.
> >
> > However, I believe my main concerns about your proposed methodology have not been clearly addressed. Specifically, my primary concern is that both the focused and fused calibrations are based solely on a subset of the data, which may lead to wasted information.
> >
> > For example, the focused calibration approach is built on the subset of the data where $e_i=1$, i.e., the portion of the data where the survival time is observed. The idea is that if there is censoring, simply drop it. This approach is likely to be highly sensitive to high censoring rates, or/and unbalanced datasets where the $e_i=1$ group is a minority. Then, the focused calibration may not real calibrate the naïve LPB. Mathematically, this sensitivity is reflected in the oracle weight, which may take on extreme values. Furthermore, the estimation of $s_{\tau}$ and the selection rules in the fused approach could also be affected. I believe a clear analysis of this issue is essential.
> > The fused calibration approach relies on data where either $e_i=1$ or $P(\tilde{T}<\hat{q}_{\tau}(X)|X=x, e=0)$
> >
> > $ < P(\tilde{T}<\hat{q}_{\tau}(X)|X=x, e=1) $.
> >
> > I doubt how often this condition $P(\tilde{T}<\hat{q}_{\tau}(X)|X=x, e=0)$
> >
> > $ < P(\tilde{T}<\hat{q}_{\tau}(X)|X=x, e=0) $
> >
> > would occur, since I would expect $\alpha_{focus}(\tau) < \alpha_{naive}(\tau) $ in most cases due to early censorship or/and $T>C$ in the censoring case.
> >
> >
> > About the estimation of $s_{\tau}(x)$, I aware that this estimation will not affect your theoretical guarantee of the coverage. I still believe that a bad estimation of $s_{\tau}(x)$ will harm the efficiency of your fused calibration, and probably push your fused approach to the focused approach. How much efficiency is costed here?
> >
> > Finally, I want to clarify that I do not mean to diminish the quality of your work due to its foundation in Gui et al. (2024). Since the general idea has already been introduced in Gui et al. (2024), your work should emphasize the advantages of your approaches (which you have done) and provide a clear analysis of the limitations or issues associated with your methodology. Such a discussion would serve as a highlight of your work rather than a weakness.

---

> ### Author Response · Authors · 2024-11-26
>
> Thank you for your feedback! We are glad that the reviewer appreciates the importance of our contribution and also for confirming the correctness of our guarantees.
>
> > My primary concern is that both the focused and fused calibrations are based solely on a subset of the data, which may lead to wasted information.
>
> We must emphasize that the whole point of conformal survival analysis methods is this: by solely relying on a subset of the data, one can generate more informative LPBs while maintaining validity. Finding such a subset is the cornerstone and key novelty of Candes et al. (2023) and Gui et al. (2024). This is also our key contribution, but in contrast to prior work, we are the first to handle general right-censored data with finite sample guarantees.
>
> > Sensitivity to high censoring rates
>
> When the censoring rate is extremely high (say 95%), we mostly have $\tilde{T} = C$. In this setting, the entire survival analysis task becomes intrinsically difficult. It would be hard to estimate $T|X$, as we would barely observe the time-to-event variable $T$. That is, regardless of the calibration step, it would be hard to obtain a useful estimate of the LPB. Of course, it would also be hard to estimate the weights $w(x)$ and the selection rule $s(x)$ for our methods.
>
> We agree that in such extreme cases, it would be advisable to apply the naive method to ensure validity. We will include this discussion in the revised paper.
>
> Yet, the above extreme setup is arguably not a common use case, as reflected by our real-world experiments: the censoring rate varies between 36.6% and 86.6% for the 6 real-world datasets we explored, and the fused method consistently performed the best among the valid calibration methods. Notably, the fused method rigorously tackles the concern raised by the reviewer regarding the utilization of the data, as we detail below.
>
> > I doubt how often the condition in Proposition 3.1 (Eq. 8) is satisfied
>
> This condition rigorously reveals when the focused method would outperform the naive approach. Importantly, this condition depends on the data, i.e., it is impossible to state apriori that one method would always be better than the other. This is precisely the key challenge addressed by the fused method, which enjoys the benefits of both the naive and focused methods.
>
> Our experiments support this argument. Specifically,
>
> (1) In synthetic settings 1, 3, 4, and 6, as well as real-world datasets TCGA, churn, and NACD, we can see that the focused method outperforms the naive approach. This indicates that the condition in Eq. 8 tends to hold.
>
> (2) In synthetic setting 5, as well as real-world datasets SUPPORT, METABRIC, and GBSG, it is evident that the naive approach outperforms the focused method. This indicates that the condition in Eq. 8 tends to be violated.
>
> (3) In all the synthetic and real-world experiments, the fused method performs as well as the best among the naive or the focused methods. Further, in synthetic settings 2, 3, and 6, the fused outperforms both the naive and focused methods. This highlights the importance of carefully selecting the subset of calibration points.
>
> > The effect of poor estimation of $s(x)$ on statistical efficiency
>
> We agree. A highly inaccurate estimation of $s(x)$ would prevent us from fully exploiting the potential of the fused calibration method. A rigorous analysis of the effect of inaccurate estimation of $s(x)$ is highly challenging not only because it involves an analysis of the training process, but also because this analysis depends on the distribution of the data. We hope to conduct such an analysis in future work, but we must stress that this is very challenging.
>
> Meanwhile, we emphasize that in our experiments we fit a random forest model with the same hyperparameters for all datasets (both real and synthetic). Despite this simple design choice, we observe a very nice gain of the fused method compared to the naive and focused methods. Specifically,
>
> (1) The GLOBAL RESPONSE: EXPERIMENTS (Robustness analysis to extreme censorship rates) demonstrates that the fused method provides a statistical gain compared to both the naive and focused methods over a wide spectrum of censoring rates, between 20% and 80%.
>
> (2) The ablation study in Section D of the supplementary “rebuttal_experiments.pdf”, demonstrates the robustness of our approach to the choice of the random forest’s hyperparameters.
>
> Thank you once again for your insightful comments. We absolutely agree with the reviewer that it is important to discuss the limitations of our methods. We will expand the current discussion in the manuscript regarding the estimation of the weights: we will include a discussion on the effect of extreme weights, strategies to mitigate this problem, and also emphasize that it is preferred to use the naive method in cases where the censorship rate is extremely high. We will also explain that a poor estimation of $s(x)$ can deteriorate statistical efficiency.

---

### Official Review · Reviewer_ovAp · 2024-11-02

**Soundness:** 4
**Presentation:** 4
**Contribution:** 4
**Rating:** 8
**Confidence:** 4

**Summary:**

The manuscript is well-structured, offering solid theoretical support and a thorough experimental evaluation. The authors present a framework to quantify predictive uncertainty in survival analysis with general right-censoring, specifically addressing cases where the censoring time is not directly observed. The main contribution lies in adapting conformal prediction methods to this broader right-censoring context, providing a lower predictive bound (LPB) for patient survival times. Unlike previous approaches that assume known censoring times, this work extends the methodology to general right-censored data, where either the censoring or survival time is observed, but not both.

**Strengths:**

- The paper extends conformal prediction methods to general right-censored data, addressing a significant gap where censoring times are unknown.
- The framework is backed by strong theoretical support, providing distribution-free, finite-sample guarantees for lower predictive bounds.
- The authors provide both simulated and real-world data experiments, showcasing the method’s validity and practical effectiveness.

**Weaknesses:**

- In Section 4.1, the proposed methods appear to underperform compared to the naive method in Setting 6 regarding the lower predictive bound (LPB). Could the authors provide additional insights into why the naive method performs exceptionally well in this specific setting?

**Questions:**

- This is a general question. Why do researchers tend to prioritize the lower predictive bound rather than the upper predictive bound for survival time? Is there any existing work on the upper predictive bound for survival time?

---

> ### Author Response · Authors · 2024-11-20
> **Response to Reviewer ovAp**
>
> We very much appreciate your positive feedback and interest in our work. In what follows, we address your comments in detail.
>
> > Comparable performance in Setting 6 for all methods
>
> Interesting point! Recall that in Section 3.1, we formulate the condition for which the naive method will outperform the focused approach, which gives rise to our proposal to enhance efficiency by estimating the ideal selection rule function $s(x)$. Importantly, if the estimate $\hat{s}(x)$ of $s(x)$ is accurate, our fused method should enjoy the benefits of both the focused and naive approaches.
> In the synthetic data experiment, the fused method generates the most informative LPBs for all settings, except for Setting 6, where the performance of the fused and focused are comparable to that of the naive approach; perhaps with a slight gain for the naive (it’s hard to make a sharp conclusion as the whiskers of all methods are overlapping). In light of the above explanation, we explain this discrepancy by the inaccurate estimation $\hat{s}(x)$ of the ideal selection rule $s(x)$. We will add this discussion to the revised manuscript.
>
> > Upper predictive bounds in survival analysis
>
> Thank you for raising this insightful question. Researchers often prioritize the lower predictive bound for survival time because it tends to provide more actionable insights in practice. For instance, knowing a minimum guaranteed survival time is particularly useful in clinical decision-making.
> Exploring the upper predictive bound is an excellent research direction with potential applications, such as in treatment optimization and automatic assignment. However, this approach faces significant challenges. One practical issue is that in datasets with substantial censorship, higher quantile estimators often lack sufficient data to train effectively. For example, estimating the 95th percentile survival time in a dataset with 10% censorship is inherently difficult. As a result, the calibration process might yield uninformative, overly large upper bounds.
> To our knowledge, there has been limited work explicitly addressing upper predictive bounds in survival analysis until very recently. The method introduced by Holmes et al. (2024) generates upper bounds with finite-sample guarantees, though it was published after our work.
> We agree this is a promising area and will discuss the challenges and opportunities for two-sided bounds in the final version of the paper.
>
> **References**
>
> Chris Holmes, and Ariane Marandon. Two-sided conformalized survival analysis. arXiv:2410.24136, 2024.

---

### Official Review · Reviewer_rRgu · 2024-11-02

**Soundness:** 3
**Presentation:** 4
**Contribution:** 3
**Rating:** 8
**Confidence:** 4

**Summary:**

This paper extends on Gui et al.'s paper (Conformalized survival analysis with adaptive cut-offs), to construct valid and reliable lower predictive bound for survival analysis in the more general right-censoring settings, where either censoring time or event time can be observed but not both. The paper provides distribution-free finite-sample guarantees under some necessary conditions, and provides some experiments to demonstrate the effectiveness.

**Strengths:**

This paper is well-written and a pleasure to read. It tackles the important topic of achieving calibrated (probably approximately correct) predictions in survival analysis with the right censoring settings, an area where previous research has not provided methods with finite-sample distribution-free guarantees. The authors try to fill this gap by utilizing the concept of covariate shift. The paper is well-motivated and introduces three methods to address the problem. Starting with the simplest approach, it gradually highlights the limitations of each method and proposes solutions to overcome them. The authors provide finite-sample guarantees for the fused method, and the results on synthetic datasets support the theoretical findings.

**Weaknesses:**

Despite all the merits I mentioned, I have concerns regarding the experimental results. The experiments are limited to oversimplified synthetic datasets and a single small real-world dataset, without comparisons to existing methods. This raises questions about the practical effectiveness and generalizability of the proposed approach.

Below are my major concerns with the paper:
1. **Lack of Comparison with Existing Methods:**
The paper does not compare the proposed method with any existing approaches. Although it is the first to provide finite-sample guarantees, this does not inherently mean it outperforms methods without such guarantees. Previous studies, such as Cresswell et al. [1], have shown that methods lacking finite-sample guarantees can perform equally well or better in practice. It is crucial to include comparative analyses with methods like those proposed by Qin et al., Meixide et al., and Qi et al. [2] to compare the empirical performance of the proposed method.
2. **Simplistic Synthetic Datasets:** The main claim relies solely on 6 synthetic datasets with simplistic settings where survival and censoring times follow lognormal and exponential distributions. These settings are not reflective of real-world scenarios. For example, event and censor distributions can be non-parametric and censoring can stem from various sources (e.g., a combination of loss to follow-up, competing risks, and administrative censoring). I recommend designing more complex synthetic datasets that better mimic the complexities encountered in practice.
3. **Inadequate Evaluation on Real-World Data:**
The evaluation of the real-world dataset is insufficient. The TCGA dataset used is small (n=1044), and the chosen evaluation metrics do not adequately support the paper's main findings. There are established metrics for assessing coverage and calibration in censored datasets, such as the distribution calibration metric discussed in Haider et al. [3]. The distribution calibration is proved to be asymptotically unbiased under the conditional independent assumption (Assumption 1.1 in this paper). Although it is not perfect (as it doesn't have a finite sample guarantee), incorporating such metrics would strengthen the validity and reliability of the experimental results.
4. **Sensitivity to High Censoring Rates:**
The proposed method employs inverse probability weighting, which is known to have high variance and be sensitive to the number of uncensored subjects, especially in datasets with high censoring rates. The paper should report the censoring rates in both synthetic and real datasets and assess the method's performance across varying levels of censoring. Additionally, testing on larger datasets would provide more robust evidence of the method's practical applicability.
5. **Challenges with Double Robustness in Practice:** While the double robustness property is theoretically appealing, achieving either condition in practical settings is challenging. The paper should discuss the practical implications of this limitation and explore potential strategies to mitigate its impact.

---

Below are my additional thoughts on the lower predictive bound (LPB) and marginal coverage/calibration:

Achieving a Probably Approximately Correct (PAC) lower predictive bound is important, but focusing solely on calibration or coverage is insufficient. Simple methods like the Kaplan-Meier estimator can provide calibrated predictions without the need for complex modeling. For example, a PAC-type LPB at 90% can be contained by finding the time when the KM curve drops below 90%.  A more important question is: how does the proposed method affect (or possibly decrease) other performance metrics by achieving a better calibration/coverage? Some important metrics include ranking accuracy (Concordance index), probabilistic accuracy (Brier score), and time-to-event prediction accuracy (mean absolute/squared error).

The requirement of a separate calibration set reduces the number of samples available for training, which could negatively impact predictive performance. Qi et al. [2] proposed a conformal-based method that addresses this issue by enhancing calibration without sacrificing other metrics. Discussing such alternatives would provide valuable context and highlight potential limitations for the current manuscript for improvement.

Moreover, providing only the LPB may not offer sufficient information for real-world applications compared to generating survival or hazard functions (which are the standard prediction outcomes for survival analysis models). These functions provide a comprehensive view of the time-to-event distribution, which is crucial for decision-making in clinical and other applied settings. LPB is a degenerated output that can be obtained from the survival function while losing a lot of other information.


[1] Cresswell et al., Conformal Prediction Sets Improve Human Decision Making. ICML 2024

[2] Qi et al., Conformalized Survival Distributions: A Generic Post-Process to Increase Calibration. ICML 2024

[3] Haider et al., Effective Ways to Build and Evaluate Individual Survival Distributions. JMLR 2020


---
**Update after rebuttal**
Thank you for your comprehensive rebuttal and the additional experimental results on complex synthetic datasets, real-world datasets, and supplementary baselines. Your responses have effectively addressed my concerns. I find the current draft to be a strong paper and am now more inclined toward acceptance. Accordingly, I am revising my rating to 8.

**Questions:**

Lines 294-295 state "inaccurate estimation of $\hat{s}_\tau$ does not impact the validity of the calibration procedure", what do you mean by that exactly? Theorem 3.1 states that the calibration is achieved with *accurate weights*, which seems to contradict the first statement.

---

> ### Author Response · Authors · 2024-11-20
> **Response to Reviewer rRgu - part 1/2**
>
> Thank you for your positive feedback! We appreciate your constructive, detailed, and valuable suggestions. In what follows, we address your concerns in detail.
>
> ## Weaknesses
>
> > Simplistic synthetic datasets
>
> Thank you for this great suggestion! We revised our synthetic experiments and generated more realistic and challenging data, with a complex conditional distribution $(T, C) | X$, early censoring, early time-to-event occurrences, and end-of-trial censorship. We believe these better mimic the complexities encountered in practical use cases.
> Please refer to GLOBAL RESPONSE: EXPERIMENTS for more details and results.
>
> > Comparison with other methods
>
> Initially, we did not compare our methods to others as no existing techniques provide finite-sample coverage guarantees in the general right-censoring setup. We agree with the reviewer that such a comparison can shed light on the importance of our contribution, especially if the uncalibrated model or an asymptotic calibration method would not obtain the required coverage in practice.
> In response, we now include comparisons to the uncalibrated model and the CSD method by Qi et al. (2024). We did not compare with the asymptotic methods by Qin et al. (2024) and by Meixide et al. (2024) as neither provides software implementations. Additionally, these contributions are concurrent with ours according to ICLR guidelines.
> We kindly refer the reviewer to our GLOBAL RESPONSE: EXPERIMENTS, showing that for some synthetic datasets, the uncalibrated model and asymptotic CSD produce LPBs that undercover the test time-to-event variables. This stands in contrast with our methods, which always attain valid coverage.
> Lastly, we note that our focus on finite-sample guarantees is motivated by applications where small sample sizes are common. This is exemplified by the real-world breast cancer TCGA dataset used in our experiments. To clarify, we are not arguing that methods without finite-sample guarantees will fail in such cases, and we acknowledge the broad applicability and great utility of asymptotic approaches. However, we view finite-sample validity as an additional safeguard. We believe our numerical experiments support this perspective, demonstrating that the asymptotic method can fail to achieve valid coverage in certain settings.
> We thank the reviewer for this valuable comment, which we believe improves our paper.
>
> > Evaluation on one real-world dataset is insufficient
>
> We refer the reviewer to the GLOBAL RESPONSE: EXPERIMENTS for new real-world data experiments, comparison to baseline methods, calibration metrics, and more.
>
> > Evaluating distribution calibration metrics
>
> We very much appreciate your suggestion. We now include the following metrics in our experiments: c-index, integrated brier score (IBS), and distribution calibration (D-Cal).
> The results for the synthetic datasets are summarized in Figure 2  in the supplementary “rebuttal_experiments.pdf”:
>
> - **c-index**: our calibration methods do not affect the c-index in practice. Further, all methods demonstrate the same performance under this measure.
>
> - **IBS and D-Cal**: we agree that IBS and D-Cal are valuable metrics for evaluating the extent to which a survival analysis model is calibrated. However, these measures can penalize methods that generate valid but conservative LPBs. Indeed, following Figure 2 in the supplementary PDF, one can see that the baseline methods (uncalibrated model and CSD) outperform our methods in terms of IBS and D-Cal, **however the baselines do not achieve the correct coverage level—see, e.g., settings 1,3,4.** This discussion underscores that the IBS and D-Cal metrics are not perfectly aligned with our goal of evaluating whether a survival analysis model provides valid LPBs with a coverage rate higher than the desired level. In fact, these metrics might suggest poor performance for methods that actually provide valid but conservative LPBs. Following the above, we believe that the development of a method that performs well under these calibration metrics while attaining valid LPBs is a promising future direction.
>
> Unfortunately, we cannot analyze the correlation between coverage and IBS/D-Cal for real-world datasets, as it’s impossible to compute the coverage. However, as shown in Figure 4 of the supplementary “rebuttal_experiments.pdf,” a similar trend emerges: e.g., the naive method performs worse in terms of IBS and D-Cal compared to the uncalibrated model. This occurs although we have reasons to believe that the naive method yields valid but conservative LPBs. Lastly, we point out that in terms of c-Index, all the methods perform the same on real-world datasets.
>
> **References**
>
> Qi et al. Conformalized Survival Distributions: A Generic Post-Process to Increase Calibration. ICLR, 2024.
>
> Qin et al. Conformal predictive intervals in survival analysis: a re-sampling approach. arXiv:2408.06539, 2024.
>
> Meixide et al. Uncertainty quantification for intervals. arXiv:2408.16381, 2024.

---

> ### Author Response · Authors · 2024-11-20
> **Response to Reviewer rRgu - part 2/2**
>
> > LPB is a degenerated output compared to the entire distribution
>
> Our approach aligns with the work of Candes et al. (2023) and Gui et al. (2024), among others, which focus on LPB estimation. However, we recognize that there are scenarios where distributional estimation may be preferred over LPB estimation alone. In such cases, we can utilize our method to calibrate multiple quantiles in a manner similar to the CSD approach by Qi et al. (2024), achieving a PAC-type LPB estimation of the entire survival distribution. In fact, we implemented this approach for measuring our methods’ calibration metrics in Figures 2 and 4 in the supplementary “rebuttal_experiments.pdf.”
>
> > Sensitivity to high censoring rates: assess the method's performance across varying levels of censoring
>
> The theoretical bounds provided in Section 3.3 are valid regardless of the censoring rate. Please refer to GLOBAL RESPONSE: EXPERIMENTS (Robustness analysis to extreme censorship rates) for an experiment that validates our argument.
>
> > Include censoring rates in both synthetic and real datasets
>
> Thank you for this constructive comment. Please refer to Tables 2 and 5 in the supplementary “rebuttel_experiments.pdf,” summarizing the censorship rates for the datasets used in the new experiments. Notably, both the synthetic and real-world datasets cover a broad range of censorship rates, ensuring comprehensive evaluation.
>
> > Double robustness in practice
>
> Great comment! Providing an accurate characterization of the effect of inaccurate estimates on the coverage rate could be an exciting future direction to explore. Our theoretical analysis in Section 3.3 provides an initial step towards this ambitious goal. We remark that Theorem 3.1 and Theorem 3.2 require that either $q$ is well estimated or $w$ is well estimated.
> The exploration of potential strategies to mitigate the impact of inaccurate estimation of the weights is intriguing. For example, one approach to improve the stability to inaccurate estimations is to avoid extreme weights. This can be done by bounding/clipping the estimated weights, as suggested by Candes et al. (2023). In the interest of space, we kindly refer the reviewer to our response to Reviewer bciN (extreme weights) for further discussion on this topic.
>
> ## Questions
>
> > Why does an inaccurate $\hat{s}_{\tau}$ not affect the coverage guarantee
>
> We thank the reviewer for allowing us to clarify this point. Recall that the weights of the fused method are defined as follows: $w(x)=1/P[e=1 | X=x]$ if $\hat{s}(x)=0$ and $1$ otherwise. Notice that this condition does not involve the oracle selection rule $s(x)$ !! Instead, it relies solely on its estimate $\hat{s}(x)$. As a result, even a poorly estimated $\hat{s}(x)$ of the oracle $s(x)$ ensures valid coverage. The intuition here is that the goal of $s(x)$ is to enhance statistical efficiency, and so the consequence of relying on its poor estimate is reduced efficiency, not invalid coverage.
> We will clarify this point in the text.
>
> **References**
>
> Candes et al. Conformalized survival analysis. JRSS Series B: Statistical Methodology, 85(1):24–45, 2023.
>
> Gui et al. Conformalized survival analysis with adaptive cut-offs. Biometrika, 111(2):459–477, 2024.
>
> Qi et al. Conformalized Survival Distributions: A Generic Post-Process to Increase Calibration. ICLR, 2024.

---

> > ### Comment · Reviewer_rRgu · 2024-11-21
> >
> > Thank you for your comprehensive rebuttal and the additional experimental results on complex synthetic datasets, real-world datasets, and supplementary baselines. Your responses have effectively addressed my concerns. I find the current draft to be a strong paper and am now more inclined toward acceptance. Accordingly, I am revising my rating to 8.
> >
> > ---
> > I would also like to address `reviewer bciN`'s comment that "the general right-censoring is quite similar to the Type-I right-censoring setting." I respectfully disagree with this assessment. As the authors correctly noted in their rebuttal, general right-censoring provides either event or censoring time, but not both, whereas Type-I censoring includes both. This fundamental difference renders many Type-I censoring-specific techniques, particularly those requiring censoring distribution estimation, inapplicable to general right-censoring settings. In my experience, the vast majority of survival datasets (though this is based on my subjective observation rather than rigorous statistical analysis) follow the general right-censoring pattern. This prevalence underscores the significance of developing conformal methods suitable for such widespread applications.

---

> > > ### Author Response · Authors · 2024-11-22
> > >
> > > Thank you for your engagement and for increasing the score! We deeply appreciate your thoughtful comments, which highlight our contributions, and your insightful suggestions, which have significantly enhanced our work.

---

### Official Review · Reviewer_R8TU · 2024-11-03

**Soundness:** 2
**Presentation:** 2
**Contribution:** 1
**Rating:** 3
**Confidence:** 3

**Summary:**

This paper introduces an extended conformal survival analysis framework specifically designed to handle general right-censored data, aiming to provide a reliable lower predictive bound (LPB) for patient survival times. The main contribution of the paper is the development of a novel, doubly robust calibration method that constructs valid LPBs with finite-sample guarantees, even under uncertain censoring conditions.

**Strengths:**

The paper proposes a doubly robust calibration method capable of constructing valid lower predictive bounds (LPBs) for general right-censored data, addressing a gap in survival analysis. The proposed method provides finite-sample guarantees, which are particularly important for ensuring model reliability in high-stakes survival analysis applications.

**Weaknesses:**

Although the paper introduces a doubly robust calibration method, it fundamentally relies on existing conformal prediction methods (e.g., the work by Gui et al., 2024). The real innovation lies in extending these techniques to the general right-censored data setting, representing a relatively incremental contribution. The comparisons in the paper are primarily against a relatively simple "naive" calibration method rather than other state-of-the-art survival analysis methods.

**Questions:**

1. In the "focused calibration" and "fused calibration" methods, the estimation of weights w(x) is crucial. How much impact does error in weight estimation have on the prediction results? Are there established standards for evaluating the accuracy of weight estimation, or strategies to reduce its sensitivity?

2. In the experiments, the proposed method is compared with the "naive method." Are there other commonly used methods for handling censored data that could serve as benchmarks? For example, traditional weighted Cox regression models or other survival analysis methods.

3. What is the computational complexity of the proposed "fused calibration" method when handling large-scale datasets in practice? How does its computational efficiency compare to other methods?

4. The proposed method relies on the estimation of weights and conditional quantiles; however, the article does not include a sensitivity analysis of these critical hyperparameters. How do the hyperparameters impact model performance? Is it possible to provide a robust hyperparameter selection strategy?

5. The effectiveness of doubly robust calibration depends on the accuracy of conditional quantile or weight estimation. Could explicit theoretical bounds be provided to describe the calibration method’s performance under inaccurate estimations? It is recommended to clearly specify how such biases impact the final coverage.

---

> ### Author Response · Authors · 2024-11-20
> **Response to Reviewer R8TU**
>
> We appreciate your constructive feedback and suggestions. In what follows, we address your concerns in detail.
>
> > The novelty of the proposed methods
>
> See GLOBAL RESPONSE: CONTRIBUTIONS.
>
> > Comparison to other methods
>
> See GLOBAL RESPONSE: EXPERIMENTS.
>
> > Effect of inaccurate weights
>
> When relying on inaccurate weights, one might obtain either under-coverage or over-coverage in practice. Our experience suggests that our proposed methods are fairly robust to inaccurate weights. This observation is in line with related conformal methods that theoretically require accurate weights for valid coverage, but they often achieve the desired level in practice (Tibshirani et al. (2019); Candes et al. (2023); and Gui et al. (2024)).
>
> From a theoretical perspective, Theorem 3.1 establishes a lower bound on the attained coverage rate based on the magnitude of the estimation error of the weights. This quantifies the worst-case drop in coverage stemming from inaccurate weight estimation.
>
> > Evaluating the accuracy of the weights, reducing sensitivity to errors
>
> The reviewer’s comment is of great interest, and it touches upon the broader context of weighted conformal prediction and other statistical methods that leverage weights to address covariate or label shift (Tibshirani et al. (2019); Candes et al. (2023); Gui et al. (2024)).
>
> Unfortunately, assessing the accuracy of the estimated weights is challenging, as it is equivalent to estimating the ground truth weights $w(x) = 1/P[e=1 \mid X]$, which is generally infeasible.
>
> As for strategies to reduce the sensitivity to estimation errors, one approach to improve stability is to avoid extreme weights. This can be done by bounding/clipping the estimated weights, as suggested by Candes et al. (2023). In the interest of space, we kindly refer the reviewer to our response to Reviewer bciN (extreme weights) for further discussion on this topic.
>
> > The computational complexity of the proposed methods
>
> All methods, including ours, require training a base predictive model, $\hat{q}$. The focused and fused calibration methods require fitting a classifier for weight estimation. The fused calibration method requires training an additional classifier $s(x)$ for each threshold considered. Training these additional classifiers can be done in parallel, and the computational complexity is comparable to that of fitting the base predictive model.
>
> As for the complexity of the calibration process, the naive, focused, and fused calibration schemes involve a linear pass over the calibration samples to estimate the miscoverage rate for a given threshold. This calibration process should only be applied once (offline), and it is computationally negligible in comparison to training the survival analysis model, for example. The inference time is identical to running the base predictive model, as it amounts to invoking the model with a calibrated quantile level. We will further elaborate and clarify this point in Appendix D.2.
>
> > Weights and conditional quantile estimation: the effect of hyperparameters
>
> In our experiments, we use a single set of hyperparameters, as detailed in Appendix D, and we apply early stopping on a validation set to avoid overfitting. Despite using such a simple strategy, our methods always attained valid coverage.
> In response to the reviewer’s comment, we conducted an ablation study to analyze the impact of key hyperparameters on model performance, specifically in estimating weights and conditional quantiles, with a focus on their influence on coverage and the resulting LPBs.
>
> For weight estimation, we test the performance of our methods using either a shallow random forest classifier (max depth 2) or a deeper one (max depth 6). For conditional quantile estimation, we test the methods using either a shallow network (1 hidden layer) or a deeper network (3 hidden layers). The results are presented in Figure 6 in the “rebuttel_experiments.pdf.” Following that figure, we can see that our calibration methods always attain valid coverage, highlighting the robustness of our methods to different design choices of the underlying predictive models.
>
> > Theoretical bounds for coverage under inaccurate estimations
>
> The impact of inaccuracies in weight and conditional quantile estimation is characterized by the error terms of Theorems 3.1 and 3.2. Theorem 3.1 shows that the coverage rate is affected by the ratio $ \hat w_\tau (x) / w_\tau (x) $ , quantifying how well the weights are estimated. Theorem 3.2 shows that conditional coverage is affected by the difference $ \hat q_{\tau(\beta)}(x) - q_{\tau(\beta)}(x) $, quantifying how well the conditional quantiles are estimated.
>
> **References**
>
> Tibshirani et al. Conformal prediction under covariate shift. NeurIPS 2019.
>
> Candes et al. Conformalized survival analysis. JRSS Series B: Statistical Methodology, 85(1):24–45, 2023.
>
> Gui et al. Conformalized survival analysis with adaptive cut-offs. Biometrika, 111(2):459–477, 2024.

---

### Author Response · Authors · 2024-11-20
**GLOBAL RESPONSE: EXPERIMENTS**

The main criticism raised was about the relatively simple synthetic experiments, limited comparison to baseline methods, and limited real-world data experiments. In response, we will revise the experimental section and include the following:

(1) More realistic synthetic datasets: the synthetic datasets now include a more complex relationship of $(T,C) | X$, early censoring, early time-to-event, and end-of-trial censorship.

(2) We now include the results of the uncalibrated base model: a deepsurv model, which is a Cox model with a deep learning model instead of a linear one.

(3) We also include the conformalized survival distributions (CSD) method by Qi et al. (2024) for right censored data—an asymptotic calibration method.

(4) We have added 5 more real-world data sets, and compared the performance of our method to the uncalibrated base model and CSD methods.

(5) We conducted a robustness analysis of our methods to extreme censorship rates.

All these experiments, and additional ones that are more reviewer-specific, appear in a PDF titled “rebuttal_experients.pdf” attached to the Supplementary Material. We will include all the new experiments in the text; in the interest of space, we will migrate the synthetic experiments of the initial version of the paper to the appendix.

**Remark:** The supplementary PDF includes additional calibration metrics (c-index, IBS, and D-Cal) suggested by Reviewer rRgu. While these metrics are helpful for assessing distribution calibration, our experiments show that these metrics are not necessarily indicative of valid but conservative coverage. In the interest of space, we are not discussing this here and kindly refer to our response to Reviewer rRgu (Evaluating distribution calibration metrics) for details.

> Results: synthetic data

In all experiments, we fit a deepsurv model on 200 training samples, apply early stopping using a validation set to avoid overfitting, and have 1000 samples reserved for calibration. The precise description of the synthetic datasets is in Section A.1 of the supplementary “rebuttal_experients.pdf.”

Following Figure 1 in the supplementary PDF, we can see that our methods (naive, focused, fused) always attain valid coverage, as expected. Our advertised method—the fused approach—outperforms the naive and focused methods in terms of relative LPB (higher is better). In contrast to our calibration methods, the uncalibrated model does not attain valid coverage in all cases. Similarly, the asymptotic CSD method does not achieve valid coverage in all cases.


> Results: real data

In addition to the breast cancer dataset (‘TCGA’), we now added the ‘SUPPORT’ (n=8873), ‘METABRIC’ (n=1904), ‘GBSG’ (n=686), ‘NACD’ (n=2401), and ‘churn’ (n=2000) datasets available at github.com/shi-ang/MakeSurvivalCalibratedAgain and github.com/havakv/pycox. Refer to Qi et al. (2024) for more information about these datasets.

We follow the same training and calibration protocol as in the synthetic experiments, except that here, we randomly split the available data into disjoint training (60% of the data), validation (10%), calibration (20%), and testing (10%) sets.

The results are summarized in Figure 3 in the supplementary “rebuttal_experients.pdf.” In contrast with the synthetic experiments, it is impossible to evaluate the coverage in such a real-world setting. With this in mind, we can see that the fused method performs the best among the three calibration methods we proposed, which are supported by finite-sample coverage guarantees. A conclusion that is in line with our synthetic experiments. The baseline methods—uncalibrated model and CSD—often attain higher LPBs, but these might be invalid; recall the synthetic experiments where these baseline methods do not always achieve the desired coverage level in practice.

> Results: Robustness analysis to extreme censorship rates

We evaluate the performance of our calibration methods for varying levels of censorship, using the dataset from Setting 3 of our new synthetic experiments. We vary the censorship rate by changing the end-of-trial time.

We follow the same experimental protocol as in the synthetic experiment, and present the results in Figure 5 in the supplementary “rebuttal_experients.pdf.” As depicted in that figure, we can see that all the methods attain valid coverage regardless of the censorship rate, as expected. The fused approach outperforms the other methods, providing valid LPBs that are closer to the desired coverage level. The focused method performs similarly to the fused method, only under a low censorship rate, and the naive method is more conservative than the others.


**References**

Shi-Ang Qi, Yakun Yu, and Russell Greiner. Conformalized Survival Distributions: A Generic Post-Process to Increase Calibration. ICML, 2024.

---

### Author Response · Authors · 2024-11-20
**GLOBAL RESPONSE: CONTRIBUTIONS**

We appreciate the reviewers' engagement with our paper and their valuable comments and suggestions. We will integrate their feedback into the revised paper.

We are glad that reviewers found our paper to be “well-structured” (Reviewer ovAp) and “well-written and pleasure to read” (Reviewer rRgu). We are also gratified that Reviewers ovAp and rRgu recognize the importance of our contribution and its novelty. To quote Reviewer rRgu: “[This paper] tackles the important topic of achieving calibrated (probably approximately correct) predictions in survival analysis with the right censoring settings, an area where previous research has not provided methods with finite-sample distribution-free guarantees.“ Thank you!

We also appreciate the concerns that Reviewers R8TU and bciN raised about the novelty of our method, and thank them for providing us the opportunity to elaborate both on the importance of our work and its key technical novelty.

> Importance of reliable survival analysis under general right-censored data

General right-censoring is a very common setup in survival analysis, where the observed data include either the event time $T$ or the censoring time $C$, but not both. We understand that this setup may be confusing, as it is often conflated with Type-I censoring. However, the Type-I setting—where the censoring time $C$ is always available—is a special case of the general right-censoring framework for which $C$ is observed only for surviving subjects. Therefore, the ability to provide reliable survival analysis for general right-censoring is crucial in many real-world scenarios, as recognized by Reviewer rRgu and Reviewer ovAp. To be sure, this setup is widely explored in the literature, see, e.g., Clark et al. (2003); Jenkins et al. (2005); George et al. (2010); Qi et al. (2024); Lagakos et al. (1979), to name a few papers.

> Key novelty

We are the first to propose conformal prediction methods that handle the broader right-censoring setting, providing a lower predictive bound (LPB) for patient survival times with finite-sample guarantees.
While our framework builds on the foundations of Gui et al. (2024), it introduces significant innovations, particularly in the design of miscoverage estimators that do not require access to censorship times $C_i$ for all samples $i=1,...,n$. This stands in striking contrast with the miscoverage estimator of Gui et al. (2024), which can only be implemented when $C_i$ is known for all $i$. To be sure, the method by Gui et al. (2024) is not applicable under general right-censored data.
The transition from the naive approach to the focused method, and then to our advertised fused method is highly non-trivial: the challenge is to carefully select a subset of observations that make the LPB more informative while rigorously ensuring the validity of our methods. We motivated and backed our design choices analytically; see, e.g., Proposition 1.
We should also stress that the theoretical validity of our proposal does not trivially follow from Gui et al. (2024), and our analysis requires additional crucial steps. Indeed, the key theoretical challenge is to prove that even though $T_i$ and $e_i$ can be dependent conditional on $X$, our calibration methods are valid. This goes beyond the technical contribution of Gui et al. (2024).

**References**

TG Clark, MJ Bradburn, SB Love, and DG Altman, Survival Analysis Part I: Basic concepts and first analyses. British Journal of Cancer, 2003.

Stephen Jenkins. Survival analysis. 2005.

Brandon George, Samantha Seals, and Inmaculada Aban. Survival analysis and regression models. Journal of Nuclear Cardiology, 2010.

Shi-Ang Qi, Yakun Yu, and Russell Greiner. Conformalized Survival Distributions: A Generic Post-Process to Increase Calibration. ICML, 2024.

SW Lagakos. General right censoring and its impact on the analysis of survival data. Biometrics, 1:139-56, 1979.

Yu Gui, Rohan Hore, Zhimei Ren, and Rina Foygel Barber. Conformalized survival analysis with adaptive cut-offs. Biometrika, 111(2):459–477, 2024.

---

### Meta-Review · Area_Chair_aRwm · 2024-12-24

**Metareview:**

The work represents a interesting advancement in survival analysis by addressing the more realistic and challenging general right-censoring scenario, where either the survival time ($T$) or the censoring time ($C$) is observed, but not both. A core novelty is the introduction of the indicator variable $e$ (denoting whether the observation is censored or uncensored), which naturally encodes whether $T$ or $C$ is observed without requiring explicit access to $C$. This is a powerful and natural idea because $e$ reflects the fundamental structure of censored data and allows the framework to handle both cases robustly without making restrictive assumptions. The work further introduces novel calibration techniques—focused calibration for uncensored data and fused calibration for selectively incorporating censored data—that ensure statistical efficiency while maintaining finite-sample, distribution-free guarantees. Its double robustness property ensures valid results if either censoring weights or survival quantile models are well-specified, making it highly reliable even under model misspecification. By leveraging $e$, improving efficiency, and expanding applicability to complex, real-world censoring patterns, this work marks a foundational step beyond incremental improvements in conformal survival analysis.

**Additional Comments On Reviewer Discussion:**

The reviewers recognized the significance of addressing general right-censoring in survival analysis, with Reviewers rRgu and ovAp praising the paper's clarity, strong theoretical guarantees, and practical importance. They emphasized that providing finite-sample distribution-free guarantees for lower predictive bounds (LPBs) under general right-censoring fills a critical gap. Reviewer rRgu specifically highlighted the thorough presentation of the methods, while ovAp commended the extensive experimental evaluation. However, Reviewer R8TU and bciN raised concerns about the novelty of the contributions, arguing that the work builds heavily on prior methods, particularly Gui et al. (2024), and introduces relatively incremental extensions. Reviewer bciN also questioned the practicality of relying on subsets of data for calibration, such as the focused and fused methods, and noted the potential inefficiency under high censoring rates. Additionally, both reviewers requested clearer comparative analyses with existing survival analysis methods and a deeper examination of limitations. In response, the authors expanded their experimental section, added comparisons with baseline methods (e.g., the CSD method by Qi et al., 2024), and clarified the theoretical contributions, emphasizing that handling general right-censoring represents a significant practical extension of existing methods. Despite these efforts, some reviewers remained skeptical about the extent of the paper's innovation, while others were convinced of its merit and its importance for real-world applications.

---

### Decision · Program_Chairs · 2025-01-22

Accept (Poster)